# Genome-Wide Characterization of DGATs and Their Expression Diversity Analysis in Response to Abiotic Stresses in *Brassica napus*

**DOI:** 10.3390/plants11091156

**Published:** 2022-04-25

**Authors:** Xiangzhen Yin, Xupeng Guo, Lizong Hu, Shuangshuang Li, Yuhong Chen, Jingqiao Wang, Richard R.-C. Wang, Chengming Fan, Zanmin Hu

**Affiliations:** 1State Key Laboratory of Plant Cell and Chromosome Engineering, Institute of Genetics and Developmental Biology, Innovation Academy for Seed Design, Chinese Academy of Sciences, Beijing 100101, China; yinxiangzhen1985@163.com (X.Y.); guoxvpeng@163.com (X.G.); hulizong@126.com (L.H.); ss.li@genetics.ac.cn (S.L.); yhchen@genetics.ac.cn (Y.C.); 2College of Advanced Agriculture Sciences, University of Chinese Academy of Sciences, Beijing 100049, China; 3College of Biology and Agriculture, Zhoukou Normal University, Zhoukou 466001, China; 4Institute of Economical Crops, Yunnan Agricultural Academy, Kunming 650205, China; jingqiao_wang@126.com; 5United States Department of Agriculture, Agricultural Research Service, Forage and Range Research Laboratory, Utah State University, Logan, UT 84322-6300, USA; richard.wang@usda.gov

**Keywords:** *Brassica napus*, DGAT, expression pattern, abiotic stresses, fatty acids

## Abstract

Triacylglycerol (TAG) is the most important storage lipid for oil plant seeds. Diacylglycerol acyltransferases (DGATs) are a key group of rate-limiting enzymes in the pathway of TAG biosynthesis. In plants, there are three types of DGATs, namely, DGAT1, DGAT2 and DGAT3. *Brassica napus*, an allotetraploid plant, is one of the most important oil plants in the world. Previous studies of *Brassica napus* DGATs (BnaDGATs) have mainly focused on BnaDGAT1s. In this study, four DGAT1s, four DGAT2s and two DGAT3s were identified and cloned from *B. napus* ZS11. The analyses of sequence identity, chromosomal location and collinearity, phylogenetic tree, exon/intron gene structures, conserved domains and motifs, and transmembrane domain (TMD) revealed that BnaDGAT1, BnaDGAT2 and BnaDGAT3 were derived from three different ancestors and shared little similarity in gene and protein structures. Overexpressing *BnaDGATs* showed that only four BnaDGAT1s can restore TAG synthesis in yeast H1246 and promote the accumulation of fatty acids in yeast H1246 and INVSc1, suggesting that the three *BnaDGAT* subfamilies had greater differentiation in function. Transcriptional analysis showed that the expression levels of *BnaDGAT1s*, *BnaDGAT2s* and *BnaDGAT3s* were different during plant development and under different stresses. In addition, analysis of fatty acid contents in roots, stems and leaves under abiotic stresses revealed that P starvation can promote the accumulation of fatty acids, but no obvious relationship was shown between the accumulation of fatty acids with the expression of *BnaDGATs* under P starvation. This study provides an extensive evaluation of BnaDGATs and a useful foundation for dissecting the functions of BnaDGATs in biochemical and physiological processes.

## 1. Introduction

Triacylglycerol (TAG), the major component of vegetable oils, consists of three fatty acids esterified to a glycerol backbone. In plants, TAG is not only mainly stored in seeds, functioning as an energy reservoir to facilitate germination and early seedling growth, but also provides precursors for membrane biosynthesis and lipid signaling, which are crucial for normal plant growth and development [1,2,3]. DGAT (diacylglycerol acyltransferase, EC2.3.1.20), the key rate-limiting enzyme of the Kennedy pathway for TAG biosynthesis, catalyzes the final and committed step in this pathway by transferring an acyl group from acyl-CoA to the *sn*-3 position of *sn*-1,2-diacylglycerol (DAG) [4,5,6]. In plants, DGATs are classified into three distinct types, namely, DGAT1, DGAT2 and DGAT3, which were first identified in *Arabidopsis* [7], castor (*Ricinus communis*) [8] and tung tree (*Vernicia fordii*) [9] and peanut (*Arachis hypogaea*) [10], respectively. Since then, many varieties of the three DGAT types have been characterized in different species, such as olive [11,12], tobacco [13], soybean [14,15,16], peanut [17,18], canola [19,20,21], tung tree [9], sunflower [22], *Tropaeolum majus* [23], maize [24] and cotton [25].

As the final enzyme in TAG biosynthesis, DGAT1s are critical to oilseed development [7,19,26] and have been highlighted as a genetic engineering target to increase storage lipid production in plants [13,27,28,29,30]. In addition, DGAT1s have been demonstrated to be associated with normal growth [30,31,32,33] and abiotic stress responses [34,35,36,37,38,39,40,41,42]. Plant DGAT1s and DGAT2s, although catalyzing the same enzyme activity, have distinct physiological functions [9,43]. Plant DGAT2s were characterized from species that produce oils enriched with unusual FAs, such as *R. communis* [8], *V. fordii* [9,43] and ironweed (*Vernonia galamensis*) [44], which produce TAGs enriched in ricinoleic acid, α-eleostearic acid and vernolic acid, respectively. *B. napus* DGAT2s (BnaDGAT2s) were shown to play an important role in the accumulation of erucic acid in *Brassica napus* in one study [45]. Some plant DGAT3s were also confirmed to be involved in TAG synthesis. For example, AhDGAT3-1 was demonstrated to be specific for 1,2-diacylglycerol rather than hexadecanol, glycerol-3-phosphate, monoacylglycerol, lysophosphatidic acid and lysophosphatidylcholine and to prefer oleoyl-CoA as an acyl donor compared to palmitoyl- and stearoyl-CoAs [10]. Heterologous expression of AhDGAT3-3 could restore TAG biosynthesis with preferential incorporation of unsaturated C18 fatty acids into lipids in yeast H1246 [17]. Similar to *Arabidopsis*, recombinant purified AtDGAT3 produced from *Escherichia coli*, although very unstable, exhibits DGAT activity in vitro [46], and transient expression of *AtDGAT3* in *Nicotiana benthamiana* confirmed its involvement in TAG synthesis and specificity towards 18:3 and 18:2 FAs [47]. Overexpression of *GhDGAT3D* improved the oil content in Arabidopsis seeds [25].

The expression patterns of *DGATs* are very helpful in analyzing their functions. In plants, most *DGAT1s* are expressed widely in different tissues (roots, stems, leaves, flowers, developing pods, seedlings, germinated seeds, etc.), especially in developing seeds [7,14,15,20,48,49]. The expression levels of *B. napus DGAT1s* (*BnaDGAT1s*) during the silique development may be different in diverse germplasm sources, and higher numbers and levels of expression of *BnaDGAT1s* are correlated with high oil germplasm [50]. The expression patterns of *DGAT2s* are diverse in different species. Olive *OeDGAT2* is highly expressed in mature and ageing tissues, for example, in the late stages of anther and ovary development [11]. Soybean *GmDGAT2D* is mainly expressed in flowers [14]. Small individual changes in the relative expression of *BnaA.DGAT2s* were discovered, but the isoforms’ expression decreased as a general trend in the late developmental stages for MAPLUS (nonedible, high-erucic acid), whereas it remained on an even level in MONOLIT (edible, low-erucic acid) [45]. Cytosolic DGAT3 was first purified and identified from developing peanut cotyledons and was highly expressed between 8 and 24 DAF (days after flowering) but not detected in leaves, roots or 30-DAF seeds [10].

Stresses have an important effect on plant *DGAT* expression. The expression levels of *DGAT1s* were increased under freezing stress conditions [41,42]. The expression of soybean *GmDGAT2D* could be promoted by cold, heat or ABA treatment but inhibited by jasmonate and insect infestation [14].

*B. napus*, one of the most important oil plants in the world, is an allotetraploid species (AACC, 2n = 38) that contains two sets of homologous chromosomes derived from its diploid progenitors, *Brassica rapa* (AA, 2n = 20) and *Brassica oleracea* (CC, 2n = 18) [51,52,53]. It was expected that there may be four copies for a gene in *B. napus*. The copy number, sequence structure, relationship between sequence and function and spatial and temporal expression patterns of *BnaDGAT1s*, *BnaDGAT2s* and *BnaDGAT3s* in *B. napus* have not been systematically analyzed. In addition, various kinds of stresses, such as P starvation, low N, drought and salinity, have important influences on the growth and development of *B. napus*, and the expression of *BnaDGATs* in response to these stresses has important value for analyzing their functions, which have not been reported.

In this study, four DGAT1s, four DGAT2s and two DGAT3s were identified and annotated widely in the *B. napus* genome. Further characterizations were performed with the analyses of sequence identity, chromosomal location and collinearity, phylogenetic tree, exon/intron gene structures, conserved domains and motifs and transmembrane domain (TMD). The TAG-synthesizing abilities were tested by overexpressing *BnaDGATs* in both yeast H1246 and INVSc1. Transcriptomic analysis and qRT-PCR were performed to detect the expression patterns of *BnaDGATs* in different tissues and under different stresses. The *cis*-elements upstream of the start codons of *BnaDGATs* were investigated by PlantCARE. Moreover, the contents of fatty acids in roots, stems and leaves under the stresses of P starvation, low N, drought and salinity were analyzed by GS-MS. This study provides an extensive evaluation of BnaDGATs and provides a useful foundation for dissecting the functions of BnaDGATs in lipid metabolism and growth under stress in *B. napus*.

## 2. Results

### 2.1. Identification and Annotation of DGAT Family Members in B. napus

The CDS and protein sequences of AtDGAT1, AtDGAT2 and AtDGAT3 from the *Arabidopsis* database TAIR (http://www.arabidopsis.org/ (accessed on 25 March 2020)) were used as queries for the BLAST search in the two *B. napus* databases (GENOSCOPE, https://www.genoscope.cns.fr/brassicanapus/ (accessed on 25 March 2020) and BnPIR, http://cbi.hzau.edu.cn/bnapus (accessed on 2 March 2021)). The results showed that three BnaDGAT1s, four BnaDGAT2s and two BnaDGAT3s in GENOSCOPE and four BnaDGAT1s, four BnaDGAT2s and two BnaDGAT3s in BnPIR were detected (Table 1). To further confirm the BnaDGAT family members, the primers (Appendix A) for cloning them were designed according to the genomic DNA and CDSs in the two *B. napus* databases, and four *BnaDGAT1s*, four *BnaDGAT2s* and two *BnaDGAT3s* (Table 1) were then cloned and identified from *B. napus* ZS11. Simultaneously, five *DGAT* genes in *Brassica rapa*, five in *Brassica oleracea*, five in *Brassica nigra* and ten in *Brassica juncea* were classified using the same BLAST search in the *Brassica* Database (BRAD, http://brassicadb.cn/#/ (accessed on 11 October 2021)), as well as eight in *B**rassica*
*carinata* in the *Brassica* Genomics Database (BGD, http://brassicadb.bio2db.com/ (accessed on 11 October 2021)) (Table 1). In addition, *DGAT* genes in other typical plants, *Arachis hypogaea*, *Glycine max*, *Ricinus communis*, *Medicago truncatula*, *Jatropha curcas*, *Oryza sativa*, *Zea mays*, *Setaria italica* and *Brachypodium distachyon*, were also detected by the same BLAST search in their corresponding genome databases (Appendix A). All sequences of the CDS, genomic DNA and protein of DGATs in *B. napus* and other typical plants in this study are presented in Appendix A.

The amino acid sequences of *BnaDGAT* genes were characterized: the protein length ranged from 317 AA to 510 AA, and the molecular weight (MW) ranged from 35.64 kDa to 57.96 kDa (Table 2). Moreover, the isoelectric point (pI) values ranged from 7.75 to 8.90, which showed that these proteins are alkaline (Table 2). Moreover, the subcellular localization signals of four BnaDGAT1s and four BnaDGAT2s were detected in the endoplasmic reticulum, as were the DGAT1s and DGAT2s of the other plants in this study (Table 2 and Appendix A). No subcellular localization signals of DGAT3s were detected in *B. napus* or other plants (Table 2 and Appendix A).

Identity analysis of BnaDGATs, BraDGATs and BolDGATs was performed by the Clustal W method based on nucleotide sequences and amino acid sequences (Appendix A). The results showed that the three DGAT subfamilies shared very low identity with each other subfamily (8.2% to 17.2%), while the similarities among eight DGAT1s, eight DGAT2s and four DGAT3s were high (72.4% to 100%) at the amino acid sequence level. The similarity between every BnaDGAT in the A subgenome and its orthologous one in the C subgenome was higher than 96%. The identity of *DGATs* based on nucleotide sequences was similar to that based on amino acid sequences (Appendix A).

### 2.2. Chromosomal Location and Collinearity Analysis

Chromosomal location (Table 2 and Appendix A) analysis showed four *BnaDGAT1s* distributed on chromosomes A07, A09, C07 and C09; four *BnaDGAT2s* distributed on chromosomes A01, A03, C01 and C07 and that two *BnaDGAT3s* were distributed on chromosomes A08 and C08. In *B. rapa*, five *BraDGATs* were located on *B. rapa* chromosomes A07, A09, A01, A03 and A08 (Appendix A) and in *B. oleracea*, five *BolDGATs* were located on *B. oleracea* chromosomes C07, C09, C01, C07 and C08 (Appendix A). Five *BnaDGAT* loci on the *B. napus* A subgenome were highly parallel with five *BraDGAT* loci on the *B. rapa* A genome, and five *BnaDGAT* loci on the *B. napus* C subgenome were highly parallel with five *BolDGAT* loci on the *B. oleracea* C genome, which suggested that *B. napus* inherited and retained all the *DGAT* genes of *B. rapa* and *B. oleracea*.

The results of gene synteny analysis (Figure 1A) showed that four *BnaDGAT1s*, four *BnaDGAT2s* and two *BnaDGAT3s* might be duplicated genes, suggesting that *BnaDGAT* genes were frequently duplicated during oilseed rape evolution. The results of comparative synteny of *DGAT* gene pairs among *A. thaliana*, *B. oleracea*, *B. rapa*, *B. napus* and *B. nigra* (Figure 1B) showed that *AtDGAT1*, *AtDGAT2* and *AtDGAT3* had collinearity relationships with two *BolDGAT1s*/two *BraDGAT1s*/four *BnaDGAT1s*/two *BniDGAT1s*, two *BolDGAT2s*/two *BraDGAT2s*/four *BnaDGAT2s*/two *BniDGAT2s* and one *BolDGAT3*/one *BraDGAT3*/two *BnaDGAT3s*/one *BniDGAT3*, respectively.

Moreover, the synonymous mutations (Ks), nonsynonymous mutations (Ka) and Ka/Ks ratios of the orthologous *DGAT* gene pairs between *B. napus* and *A. thaliana* and the paralogous *BnaDGAT* gene pairs were evaluated (Table 3). The results showed that all of the Ka/Ks ratios of these gene pairs were lower than 1 (Table 3), indicating that these gene pairs experienced strong purifying selective pressure. The duplication time of these gene pairs was presumed using the formula T = Ks/2R, with R (1.5 × 10^−8^) representing neutral substitution per site per year. The results showed that the duplication times of the orthologous *DGAT* gene pairs between *B. napus* and *A. thaliana* ranged from 10.67 MYA to 20.11 MYA (Table 3), with an average value of 15.06 MYA. These results indicated that DGATs of *B. napus* diverged from *A. thaliana* ~16 MYA, which was consistent with the recent whole-genome triplication event that occurred approximately 9–15 MYA or even 28 MYA [55]. The corresponding duplication times of the paralogous *BnaDGAT* gene pairs ranged from 1.37 to 12.78 MYA, with an average value of 8.36 MYA (Table 3). Two peaks of duplication times were observed in *B. napus*: one peak (1.37–3.57 MYA) represented the duplication time of these genes, which occurred during the divergence of the A genome and C genome of *Brassica*, and the other peak (10.95–12.78 MYA), representing a duplication time of ~10 MYA, corresponded to the *Brassica* whole-genome triplication event (9–15 MYA) [55]. Therefore, the processes of *B. napus* speciation and *Brassica* whole-genome triplication likely played important roles in the divergence of the *BnaDGAT* duplicated genes in *B. napus*.

### 2.3. Evolutionary Relationship and Exon/Intron Gene Structure Analysis of BnaDGATs

The phylogenetic tree (Figure 2A) showed that three well-supported clades were grouped as the DGAT1 subfamily, DGAT2 subfamily and DGAT3 subfamily. Within the DGAT1 subfamily and DGAT2 subfamily, four BnaDGAT1s and four BnaDGAT2s, respectively, were grouped more closely with the corresponding DGAT subfamilies of the other five *Brassica* U’s triangle species (*B. rapa*, *B. oleracea*, *B. nigra*, *B. juncea* and *B. carinata*) and *Arabidopsis* than to those of the other dicot or monocot plants. Within the DGAT3 subfamily, two BnaDGAT3s were grouped more closely to those of *B. oleracea*, *B. rapa*, *B. nigra*, *B.*
*carinata* and *Arabidopsis* than to the corresponding subfamilies of the other dicot or monocot plants. The DGAT1s, DGAT2s and DGAT3s of the four monocot plants were always grouped together, rather than with those of the twelve dicot plants. These results indicated that the duplication events of *DGAT* genes occurred after the divergence of dicot and monocot plants.

The results of the exon/intron gene structures (Figure 2B and Appendix A) revealed that the gene structures of most *DGAT1s*, *DGAT2s* and *DGAT3s* were highly and independently conserved during evolution but confirmed distinct differences among the three *DGAT* subfamilies, suggesting that the three *DGAT* subfamilies evolved separately from three different ancient genes. All four *BnaDGAT1s* have 16 exons as well as *DGAT1s* from most other plants, except *BjuB029654* (13 exons), *BjuB028615* (14 exons) and *BraA07g001370.3C* (14 exons). Most *DGAT2s* are generally composed of seven to ten exons. In addition, *BnaA01G0206700ZS* had seven exons and *BnaA03G0420700ZS* had six exons in the *B. napus* database BnPIR (Appendix A), but we cloned these genes and confirmed that they (*BnaA.DGAT2.a* and *BnaA.DGAT2.b*) had nine and eight exons, respectively. Members of *DGAT3s* from *B. napus* and the selected plants in this paper contain two exons, except *BcaC07g37153* (four exons), *BjuA014363* (six exons) and *BjuB045147* (five exons).

### 2.4. The Conserved Domains and Motif Analyses

To investigate the conserved motifs and domains, the DGAT proteins from *B. napus* and the other selected plants were predicted by both the Conserved Domain Database (CDD) with Batch CD-Search in NCBI [57] and the MEME program (https://meme-suite.org/meme/tools/meme (accessed on 21 October 2021)) (Bailey et al., 2009), as shown in Figure 2C and Appendix A. All DGAT1s contained the PLN02401 domain (diacylglycerol o-acyltransferase) or MBOAT domain (membrane-bound O-acyltransferase family). All DGAT2 homologues shared the PLN02783 domain (diacylglycerol o-acyltransferase), except AH13G38440.1 and Jcr4U29423.10, which had an LPLAT superfamily domain. All DGAT3 homologues shared the TRX_Fd_family domain, except BjuB045147 and BjuA014363.

Additionally, the top 20 conserved motifs were identified in the DGAT proteins (Appendix A). All the DGAT1 homologues shared one motif 2, one motif 3, one motif 7, one motif 8, one motif 9, one motif 12 and one motif 15. All DGAT1 homologues had one motif 1, except BjuB029654 and BjuB028615. All DGAT1 homologues had one motif 4, except BjuB029654 and 29912.m005373. All DGAT1 homologues had one motif 20, except BjuB029654 and Zm00001d036982. All DGAT1 homologues, except BjuA046403, had one motif 13. All DGAT1 homologues of six *Brassica* species, *A. thaliana*, and *A. hypogaea* shared one motif 19. All DGAT2 homologues had one motif 5, one motif 6, one motif 10, one motif 11, one motif 12 and one motif 14. All DGAT2 homologues had one motif 17, except BraA03g045590.3C, AH13G38440.1 and AH05G01580.1. All DGAT2 homologues had one motif 16, except BraA03g045590.3C and AH13G38440.1. All DGAT3 homologues had one motif 18 at their C-terminus, except BjuB045147 and BjuA014363. All DGAT3 homologues shared one or two motifs 15 at their C-terminus, except Os05g04620.1, Seita.3G054300.1 and Zm00001d024765. These results supported the hypothesis that the DGAT1, DGAT2 and DGAT3 subfamilies evolved separately during eukaryote evolution, as demonstrated by the phylogenetic tree and gene structure (Figure 2).

### 2.5. Putative Transmembrane Domains of DGAT Proteins

The putative transmembrane domains (TMDs) of DGAT proteins from *B. napus* and other plants were predicted by TMHMM (https://services.healthtech.dtu.dk/service.php?TMHMM-2.0 (accessed on 10 April 2021)) (Figure 3 and Appendix A) [58]. Each of the DGAT1s was predicted to harbor seven to ten putative TMDs. For *B. napus*, BnaA.DGAT1.a and BnaC.DGAT1.a had eight putative TMDs, and BnaA.DGAT1.b and BnaC.DGAT1.b shared nine putative TMDs. For most DGAT2s, one to four putative TMDs were detected. Each of four BnaDGAT2s had two putative TMDs at the N-terminus with a large cytosolic C-terminal domain. There were no putative TMDs in any of the examined DGAT3s, consistent with their soluble nature.

### 2.6. Oil Droplets in S. cerevisiae H1246 Overexpressing BnaDGATs

DGAT is the rate-limiting enzyme in the last step of TAG synthesis, and TAG is the main component of oil bodies. Therefore, the yeast mutant H1246 (*MATα are1-Δ::HIS3, are2-Δ::LEU2, dga1-Δ::KanMX4, lro1-Δ::TRP1 ADE2*) [59] that are defective in TAG biosynthesis were used to determine whether the ten BnaDGATs are able to complement the mutated enzymes and allow the synthesis and accumulation of TAG. As shown in Figure 4, many obvious oil bodies were detected in yeast H1246 overexpressing *BnaDGAT1s*, while no obvious oil bodies were detected in H1246 overexpressing *BnaDGAT2s* and *BnaDGAT3s*, suggesting that only the four BnaDGAT1s were able to re-establish TAG synthesis in yeast H1246.

### 2.7. Fatty Acid Profiles in Yeast H1246 and INVSc1 Expressing BnaDGATs

In this study, the fatty acids in *S. cerevisiae* H1246 and INVSc1 (*MATa his3Δ1 leu2 trp1-289 ura3-52/MATα his3Δ1 leu2 trp1-289 ura3-52*) expressing *BnaDGATs* were quantitatively analyzed by GC–MS to evaluate the effect of BnaDGATs on the accumulation of fatty acids in *B. napus* (Figure 5). As shown in Figure 5A, in H1246, the expression of *BnaDGAT1s* resulted in a slight accumulation of C10:0 and C12:0 fatty acids and a significant accumulation of C14:0, C14:1, C16:0, C16:1n7, C18:0 and C18:1n9 fatty acids and increased the content of total fatty acids by one–two times but significantly decreased the content of C18:1n7 fatty acids. The expression of *BnaDGAT2s* and *BnaDGAT3s* promoted the accumulation of both C14:0 and C14:1 fatty acids. As shown in Figure 5B, in INVSc1, the expression of *BnaDGAT1s* resulted in the significant accumulation of C10:0, C12:0, C14:0, C14:1, C16:0, C16:1n7 and C18:0 and increased the total fatty acid content by approximately 0.23- to 1-fold; the expression of *BnaA.DGAT2.a*, *BnaC.DGAT2.a*, *BnaA.DGAT2.b*, *BnaA.DGAT3* or *BnaC.DGAT3* resulted in the accumulation of C10:0, C14:0 and C14:1. By comparing the effects of *BnaDGATs* on fatty acid accumulation in yeast H1246 and INVSc1, it was found that the expression of *BnaDGAT1s* could promote the accumulation of C10:0, C12:0, C14:0, C14:1, C16:0, C16:1n7, C18:0 and total fatty acids, while the expression of *BnaDGAT2s* and *BnaDGAT3s* could not promote the accumulation of total fatty acids in H1246 and INVSc1.

### 2.8. Transcriptomic and qRT-PCR Analysis of BnaDGATs in Different Tissues

The transcriptomic expression data of the *BnaDGAT* gene family from the different tissues of *B. napus* were obtained from BnTIR (http://yanglab.hzau.edu.cn/BnTIR (accessed on 10 November 2021)) [60] and BrassicaEDB (https://brassica.biodb.org/ (accessed on 10 November 2021)) [61]. The extracted data from BnTIR and BrassicaEDB were normalized by log_2_(TPM) and log_2_(FPKM), respectively, and the heatmaps were generated by TBtools (Figure 6). The results showed that the expression patterns of the three *BnaDGAT* gene families were different among diverse tissues. Every homologous pair of *BnaA.DGAT1.a* and *BnaC.DGAT1.a*, *BnaA.DGAT1.b* and *BnaC.DGAT1.b*, *BnaA.DGAT2.a* and *BnaC.DGAT2.a*, *BnaA.DGAT2.b* and *BnaC.DGAT2.b*, and *BnaA.DGAT3* and *BnaC.DGAT3* shared a similar expression pattern. The expression levels of *BnaA.DGAT2.b* and *BnaC.DGAT2.b* were lower than those of other *BnaDGATs* in most detected tissues, while the expression levels of *BnaA.DGAT3* and *BnaC.DGAT3* were higher than those of other *BnaDGATs* in most detected tissues. The expression levels of *BnaA.DGAT1.a* and *BnaC.DGAT1.a* in seeds and embryos gradually increased during seed and embryo development. The expression levels of *BnaA.DGAT1.b* and *BnaC.DGAT1.b* in seeds first gradually increased and then gradually decreased during seed development. The expression levels of *BnaA.DGAT2.a* and *BnaC.DGAT2.a* in seeds, silique walls and embryos first gradually increased and then gradually decreased during the development of seeds, silique walls and embryos. In particular, *BnaA.DGAT1.b* and *BnaC.DGAT1.b* were highly expressed in anthers and stamens.

### 2.9. Expression Analysis of BnaDGATs under Different Stresses

Six-week-old *B. napus* seedlings were used to investigate the expression patterns of *BnaDGATs* in roots, stems and leaves under abiotic stresses, including P starvation, low N, drought and salinity, by qRT–PCR.

Under P starvation, compared with mock, the expression levels of *BnaDGAT1s*, *BnaA.DGAT2.b* and *BnaC.DGAT2.b* in roots, stems and leaves were lower; the expression patterns of *BnaA.DGAT2.a* and *BnaC.DGAT2.a* in roots, stems and leaves were not changed significantly; the expression levels of two *BnaA.DGAT3s* in leaves were decreased, while their expression levels in roots and stems were not changed significantly (Figure 7).

During low N, compared with mock, the expression patterns of four *BnaDGAT1s* in stems and leaves and *BnaA.DGAT1.b* in roots were not changed, while the expression levels of *BnaA.DGAT1.a* and *BnaC.DGAT1.a* in roots were lower. For *BnaDGAT2s*, the expression patterns of *BnaA.DGAT2.a* and *BnaC.DGAT1.a* in roots and stems under low N were not changed, while their expression levels in leaves were lower; the expression levels of *BnaA.DGAT2.b* in leaves and stems were lower, while its expression levels in roots were higher; the expression levels of *BnaC.DGAT2.b* in stems were lower, while its expression levels in leaves and roots were higher. For the two *BnaDGAT3s*, their expression patterns in roots and stems were quite consistent with those in mock, while their expression levels in leaves decreased at the late stages of low N treatment (Figure 7).

During salt stress, compared with mock, the expression levels of four *BnaDGAT1s* in the stems were higher, while their expression levels in roots were lower. For *BnaDGAT2s*, the expression levels of *BnaA.DGAT2.a* and *BnaC.DGAT2.a* in roots, stems and leaves were higher; the expression levels of *BnaA.DGAT2.b* in leaves and stems were lower, while its expression levels in roots were higher; the expression levels of *BnaC.DGAT2.b* in stems were lower, but its expression levels in roots were higher. For *BnaDGAT3s*, the expression levels of *BnaA.DGAT3* in stems and leaves and *BnaC.DGAT3* in roots, stems and leaves were lower than those in mock with 12–48 h stress (Figure 7).

During drought stress, compared with mock, the expression levels of four *BnaDGAT1s* in leaves at 12 h of treatment and *BnaC.DGAT1.a BnaA.DGAT1.b* and *BnaC.DGAT1.b* in roots at 48 h of treatment were higher, while the expression patterns of *BnaA.DGAT1.a* and *BnaC.DGAT1.a* in stems were quite consistent. For *BnaDGAT2s*, the expression levels of *BnaA.DGAT2.a* and *BnaC.DGAT2.a* in stems and leaves were higher; the expression levels of *BnaA.DGAT2.b* in leaves and stems were lower, while its expression levels in roots were higher; the expression patterns of *BnaC.DGAT2.b* in leaves and stems were quite consistent with those in mock, while its expression levels in roots were higher than those in mock. For *BnaDGAT3s*, the expression levels of *BnaA.DGAT3* in stems and roots and *BnaC.DGAT3* in roots were higher than those in mock, while the expression levels of *BnaC.DGAT3* in leaves treated for 12–48 h were quite low (Figure 7).

### 2.10. Analysis of Fatty Acids in B. napus Seedlings under P Starvation, Low N, Drought and Salinity Stresses

Under the stresses of P starvation, low N, drought and salinity, the contents of fatty acids in *B. napus* roots, stems and leaves were measured by GC–MS (Figure 8).

Under P starvation, compared with mock, the content of total fatty acids in roots at 6 h to 48 h, in stems at 3 h to 48 h and in leaves at 1 h to 48 h significantly increased, as well as the content of C18:2n6c and C18:3n3 in roots, stems and leaves at 1 h to 48 h; the content of 16:0 decreased in roots at 1 h and 3 h and in stems and in leaves at 1 h, whereas it increased in roots at 6 h, 24 h and 48 h and in stems and in leaves at 3 h to 48 h; the content of 18:0 decreased in roots at 1 h to 12 h, in stems at 1 h and 3 h and in leaves at 1 h, 3 h and 12 h, whereas it increased in stems at 12 h to 48 h and in leaves at 24 h and 48 h; the content of 18:1n7 decreased in stems at 1 h but increased in roots at 24 h, in stems at 3 h, 6 h and 24 h and in leaves at 3 h, 12 h and 24 h (Figure 8A).

Under low N stress, compared with mock, the accumulation of total fatty acids was promoted in roots at 1 h to 6 h and 24 h and in stems at 24 h and 48 h but inhibited in stems at 6 h and in leaves at 48 h; the content of 16:0 decreased in roots at 1 h to 48 h, in stems at 6 h to 48 h and in leaves at 1 h, 3 h, 12 h and 48 h; the content of 18:0 decreased in roots and stems at 1 h to 48 h and in leaves at 1 h, 3 h, 12 h and 48 h; the content of 18:1n7 decreased in roots at 3 h, 6 h, 24 h and 48 h, in stems at 6 h and 48 h and in leaves at 1 h and 6 h to 48 h; the accumulation of C18:2n6c increased in roots and stems at 1 h to 48 h but decreased in leaves at 6 h, 24 h and 48 h; the content of C18:3n3 increased in roots and stems at 1 h to 48 h and in leaves at 1 h, 6 h and 48 h (Figure 8B).

Under drought stress, compared with mock, the accumulation of total fatty acids was reduced in roots and leaves at 3 h and 6 h and in stems at 1 h to 6 h, while its accumulation in stems at 12 h and 24 h was elevated; the accumulation of 16:0 and 18:0 in roots and stems at 1 h to 48 h and in leaves at 1 h to 12 h and 48 h was reduced, as well as the accumulation of 18:1n7 in roots at 3 h, in stems at 48 h and in leaves at 12 h to 48 h; the accumulation of C18:3n3 in roots, stems and leaves at 1 h to 48 h was increased, as well as the content of C18:2n6c in roots at 1 h to 48 h and in stems at 1 h to 24 h, while the content of C18:2n6c in leaves at 3 h, 6 h, 24 h and 48 h was reduced (Figure 8C).

Under salt stress, compared with mock, the accumulation of total fatty acids was reduced in roots at 1 h to 24 h and in stems at 1 h to 6 h and 48 h, while its accumulation was elevated in leaves at 6 h and 48 h; the content of 16:0 was reduced in roots at 3 h to 24 h, in stems at 1 h to 6 h and in leaves at 1 h, 3 h and 12 h, as well as the content of 18:0 in roots at 1 h to 48 h, in stems at 1 h to 6 h and in leaves at 1 h, 3h and 12 h; the accumulation of C18:2n6c was increased in roots and stems at 1 h to 48 h and in leaves at 1 h, 3 h and 48 h, as well as the accumulation of C18:3n3 in roots at 1 h to 6 h, 24 h and 48 h, in stems at 1 h to 6 h and 48 h and in leaves at 1 h to 48 h. (Figure 8D).

### 2.11. Cis-Elements in BnaDGAT Promoters and Transcription Factors and miRNA Regulating BnaDGATs

To gain more insights into the potential function and regulatory mechanism of ten *BnaDGATs*, we analyzed the *cis*-regulatory elements in their putative promoters by using the plantCARE database. The putative promoters of ten *BnaDGATs* were cloned as 1759 bp, 1488 bp, 1657 bp, 836 bp, 775 bp, 2151 bp, 1973 bp, 1660 bp and 1459 bp from ZS11 (Table 2 and Appendix A). The putative promoters of three *AtDGATs* were derived from TAIR (Appendix A). The *cis*-acting regulatory elements in the 776–1500 bp upstream promoter regions of *BnaDGATs* and *AtDGATs* were displayed (Figure 9 and Appendix A).

As the putative promoters of *DGAT1s*, they all shared some light-responsive elements. All *P_AtDGAT1_*, *P_BnaA.DGAT1.a_*, *P_BnaC.DGAT1.a_*, and *P_BnaC.DGAT1.b_* shared anaerobic induction elements and gibberellin-responsive elements. *P_AtDGAT1_*, *P_BnaA.DGAT1.a_* and *P_BnaC.DGAT1.a_* possessed MeJA-responsive elements and defense and stress-responsive elements. Both *P_AtDGAT1_* and *P_BnaA.DGAT1.a_* contained a circadian control element. Both *P_BnaA.DGAT1.b_* and *P_BnaC.DGAT1.b_* had a low-temperature-responsive element. *P_AtDGAT1_* contained a drought-inducibility element. *P_BnaC.DGAT1.a_* contained an SA-responsive element. *P_BnaA.DGAT1.a_* contained two GCN4_motifs and three Skn-1_motifs, which are endosperm-expressive elements. In addition, *P_BnaA.DGAT1.a_* contained a seed-specific RY element. There was only one Skn-1_motif in *P_BnaC.DGAT1.a_*.

The putative promoters of *AtDGAT2* and *BnaDGAT2s* also contained some light-responsive elements, anaerobic induction elements and circadian control elements. *P_AtDGAT2_*, *P_BnaA.DGAT2.a_* and *P_BnaA.DGAT2.b_* shared a drought-responsive element. *P_AtDGAT2_*, *P_BnaA.DGAT2.b_* and *P_BnaC.DGAT2.b_* contained gibberellin-responsive elements. *P_BnaC.DGAT2.a_*, *P_BnaA.DGAT2.b_* and *P_BnaC.DGAT2.b_* shared a Skn-1_motif and an ABA-responsive element. Both *P_BnaA.DGAT2.b_* and *P_BnaC.DGAT2.b_* had a low-temperature-responsive element. Both *P_BnaA.DGAT2.a_* and *P_BnaC.DGAT2.a_* shared an elicitor-responsive element. Both *P_BnaA.DGAT2.b_* and *P_BnaC.DGAT2.b_* contained one MeJA-responsive element. *P_AtDGAT2_* contained an auxin-responsive element and an SA-responsive element. *P_BnaC.DGAT2.a_* contained a heat-stress-responsive element and a wound-responsive element.

*P_AtDGAT3_*, *P_BnaA.DGAT3_* and *P_BnaC.DGAT3_*, also contained some light-responsive elements and ABA-responsive elements. Both *P_BnaA.DGAT3_* and *P_BnaC.DGAT3_* contained MeJA-responsive elements. Both *P_AtDGAT3_* and *P_BnaA.DGAT3_* shared a defense- and stress-responsive element. Both *P_AtDGAT3_* and *P_BnaC.DGAT3_* contained low-temperature-responsive elements. *P_AtDGAT3_* contained a gibberellin-responsive element and an auxin-responsive element. *P_BnaA.DGAT3_* contained an endosperm expression element and an anaerobic induction element. *P_BnaC.DGAT3_* contained a drought-responsive element.

Transcription factors (TFs) regulate the precise initiation of gene transcription by binding the *cis*-acting elements of gene promoters. Therefore, we identified the target TFs putatively regulating the expression of *BnaDGATs* using the PlantRegMap server, and a total of 209 relationships were identified (Appendix A). Three *BnaDGAT* gene families may be regulated by different TFs, such as the B3 family, dehydration-responsive element-binding protein (DREB), GATA, ethylene response factor (ERF), WRKY family, MYB and bZIP family, indicating that these TFs regulate plant development and stress responses.

miRNAs have been widely shown to play an important role at the transcriptional and posttranscriptional levels in regulating gene expression under stress [63,64]. Therefore, we predicted that the bna-miRNAs would target *BnaDGATs* using the psRNATarget website (www.zhaolab.org/psRNATarget/ (accessed on 25 November 2021)). The results showed that four miRNAs (bna-miR2111a-3p, bna-miR390a, bna-miR390b and bna-miR390c) only targeted *BnaA.DGAT2.b* and *BnaC.DGAT2.b* and that no miRNAs targeted *BnaDGAT1s* or *BnaDGAT3s* (Appendix A). For *B. rapa*, bra-miR9562-5p was predicted to target *BraDGAT1s* (*BraA07g001370.3C* and *BraA09g011830.3C*), bra-miR2111a-3p to target one *BraDGAT2* (*BraA03g045590.3C*) and bra-miR5721 to target *BraDGAT2s* (*BraA01g022340.3C* and *BraA03g045590.3C*) (Appendix A). For *B. oleracea*, no miRNA was predicted to target the *BolDGATs*. In addition, six miRNAs (ath-miR3434-5p, ath-miR5020b, ath-miR5629, ath-miR5633, ath-miR5654-3p and ath-miR858a) were predicted to target *AtDGAT1*, six miRNAs (ath-miR2936, ath-miR390a-5p, ath-miR390b-5p, ath-miR5641, ath-miR835-5p and ath-miR847) to target *AtDGAT2* and two miRNAs (ath-miR847 and ath-miR4221) to target *AtDGAT3* (Appendix A).

## 3. Discussion

*B. napus* is an allotetraploid (AACC) crop that originated from the hybridization of two diploid progenitors, *B. rapa* (AA) and *B. oleracea* (CC) [51,52]. In this study, ten *BnaDGATs* were cloned and identified from *B. napus* and grouped into three subfamilies—BnaDGAT1, BnaDGAT2 and BnaDGAT3—based on their homology. Systematic analyses of chromosome location, gene synteny, physicochemical properties, phylogenetic tree, exon/intron gene structure, conserved domain and motif compositions, TMDs and the distribution of *cis*-elements in the promoters were performed. The functions of BnaDGATs were detected in yeast H1246 and INVSc1. Moreover, qRT–PCR and prepublished RNA-seq data were analyzed to determine the expression patterns of *BnaDGATs*. These results provide an extensive evaluation of BnaDGATs and a useful foundation for dissecting the functions of BnaDGATs in biochemical and physiological processes.

### 3.1. Gene Duplication and Functional Diversification of DGAT Family Members

The phylogenetic tree showed that BnaDGATs were grouped into three well-supported clades: the DGAT1 subfamily, the DGAT2 subfamily and the DGAT3 subfamily (Figure 2A). Identity analysis showed that the three DGAT subfamilies shared very low identity with each other subfamily (Appendix A). Gene synteny analysis showed that there was no gene synteny among the three DGAT subfamilies (Figure 1). The analysis of gene structures revealed distinct differences among the three *DGAT* subfamilies in exon/intron gene structure (Figure 2B and Appendix A). Conserved domain and motif analyses showed that the three *DGAT* subfamilies had different conserved domains and motifs (Figure 2C and Appendix A). Each DGAT1 was predicted to harbor seven to ten putative TMDs, one to four putative TMDs were detected in most DGAT2s and there were no putative TMDs in any of the examined DGAT3s (Figure 3 and Appendix A). The expression patterns of the three *BnaDGAT* gene families were different among diverse tissues (Figure 6). All of these results showed that the *DGAT1*, *DGAT2* and *DGAT3* gene subfamilies showed apparent differences and indicated that they are divergent genes and may have a distinct origin, consistent with what is described in soybeans [65] and upland cotton [25]. In addition, the results of this study showed that three *BnaDGAT* gene subfamilies were frequently duplicated during the speciation and evolution of *B. oleracea*, *B. rapa* and *B. napus* and suggested that *B. napus* inherited and retained all the *DGAT* genes of *B. rapa* and *B. oleracea* (Figure 1 and Appendix A), consistent with what is described in tetraploid cotton [25].

### 3.2. Role of BnaDGATs in Oil Biosynthesis

DGAT1 has been functionally confirmed in oil biosynthesis in Arabidopsis, soybean, oilseed rape, and so on [3]. Expression analysis revealed that DGAT1 was abundantly expressed in the developing embryos in several oilseed crops, and its transcript level was correlated with oil accumulation in developing seeds [49]. Analysis of the putative promoters showed that *P_BnaA.DGAT1.a_* contained two GCN4 motifs (endosperm expressive elements), three Skn-1 motifs (endosperm expressive elements) and one seed-specific RY-element and that *P_BnaC.DGAT1.a_* possessed one Skn-1 motif (Figure 9). In this study, the expression levels of *BnaA.DGAT1.a* and *BnaC.DGAT1.a* in seeds and embryos gradually increased during seed and embryo development (Figure 6), which corresponded with the rapid oil accumulation stage in canola seeds, indicating that BnaA.DGAT1.a and BnaC.DGAT1.a were important in TAG biosynthesis in canola seeds. The expression levels of *BnaA.DGAT1.b* and *BnaC.DGAT1.b* in seeds first gradually increased and then gradually decreased during seed development, and the expression levels of *BnaA.DGAT2.a* and *BnaC.DGAT2.a* in seeds, silique walls and embryos first gradually increased and then gradually decreased during the development of seeds, silique walls and embryos (Figure 6), which implied that BnaA.DGAT1.b, BnaC.DGAT1.b, BnaA.DGAT2.a and BnaC.DGAT2.a may also play an important role in TAG biosynthesis in canola seeds. Moreover, there were high expression levels of *BnaA.DGAT1.b* and *BnaC.DGAT1.b* in the anthers and stamens of canola, indicating that BnaDGAT1 might be involved in the reproductive development of oilseed rape.

It was reported that tung tree TAG production via DGAT1 and DGAT2 occurs in a distinct ER subdomain and that DGAT1 and DGAT2 differ in substrate preference [9]. In this study, BnaDGAT1s and BnaDGAT2s were predicted to harbor putative TMDs and to be located in the ER (Table 2 and Figure 3). Then, we overexpressed *BnaDGAT1s* and *BnaDGAT2s* in yeast H1246 and INVSc1 and found that only four *BnaDGAT1s* were able to re-establish TAG synthesis in yeast H1246 (Figure 4) and could promote the accumulation of total fatty acids in H1246 and INVSc1 (Figure 5). In previous studies, 16:0-CoA and 18:1-CoA were the best substrates of BnaA.DGAT1.a, BnaC.DGAT1.a and BnaC.DGAT1.b with [^14^C]glycerol labelled di-6:0-DAG as an acyl acceptor and with either 16:0-CoA, 18:0-CoA, 18:1-CoA, 18:2-CoA, 18:3-CoA or 22:1-CoA as acyl donors [45], and the preference of four BnaDGAT1s for 16:0-CoA was four–seven times as high as that for 18:1-CoA when the ratio of 18:1-CoA to 16:0-CoA was 1:1, while their preference for 18:1-CoA was two–five times as high as that for 16:0-CoA when the ratio of 18:1-CoA to 16:0-CoA was 3:1 [20]. However, the N-terminus BnDGAT1_(1-116)_ of BnDGAT1 (AF164434, BnaA.DGAT1.b) binds to 22:1*cis*^Δ13^-CoA more strongly than 18:1*cis*^Δ9^-CoA [66], and purified BnaC.DGAT1.a exhibited substrate preference for 18:3-CoA > 18:1-CoA = 16:0-CoA > 18:2-CoA > 18:0-CoA [67]. Here, BnaDGAT1s were found to prefer C16 (C16:0 and C16:1n7) and C18 (C18:0 and C18:1n9) as substrates instead of C10, C12 and C14 in yeast H1246 and INVSc1 (Figure 5). DGAT2s were previously found to prefer unusual or polyunsaturated fatty acids [8,9,43,44,68,69,70]. Unusual or polyunsaturated fatty acids were not detected in H1246 and INVSc1 in this study, which may be because saturated and monounsaturated fatty acids are the main components of the fatty acids in *S. cerevisiae* [71].

Some studies have focused on the role of DGAT3 in TAG biosynthesis [10,25,46,47,72]. In this study, almost all DGAT3 homologues were found to share the TRX_Fd_family domain (Figure 2C). To date, AtDGAT3 and CsDGAT3 have been confirmed as metalloproteins involved in TAG biosynthesis in plants [46,72]. DGAT3 homologues were not found in mossy or algal species [73], indicating that they may have arisen during plant evolution. The expression levels of *BnaDGAT3s* were significantly higher than those of *BnaDGAT1s* and *BnaDGAT2s* during canola seed development (Figure 6), consistent with what is described in upland cotton and soybean [65,74]. In this study, overexpressing *BnaDGAT3s* could not restore TAG synthesis in yeast H1246 and did not promote the accumulation of fatty acids (Figure 4 and Figure 5). Therefore, the function of BnaDGAT3s in synthesizing TAGs needs further testing using oilseed rape or other protein expression systems.

### 3.3. The Response of BnaDGATs to Abiotic Stresses

Phosphorus starvation can increase the accumulation of oils in most microalgae [75,76,77,78,79,80] as well as in the vegetative tissues of plants, such as *Arabidopsis*, tomato, tobacco and barnyard grass (*Echinochloa crusgalli*) [81,82]. In this study, we found that the accumulation of total fatty acids was greatly promoted (Figure 8A), but the expression levels of *BnaDGAT1s*, *BnaDGAT2s* and *BnaDGAT3s* were not enhanced in the roots, stems and leaves of *B. napus* seedlings under P starvation (Figure 7). In *Arabidopsis*, the expression of genes involved in TAG synthesis, such as *AtDGAT1*, *AtDGAT2*, *AtPDAT1*, *AROD1*, *AtLPCAT2*, *AtBCCP2* and *PDH-E1a*, was not increased under P starvation [81,82]. The above studies showed that the expression levels of *DGAT1s* were not directly related to the increase in TAG accumulation under P starvation. In previous studies, a self-inhibiting motif in the N-terminal region of BnaDGAT1 bound PA and shifted BnaDGAT1 to a higher activity state [83,84]; PLDZ2 (phospholipase D Z2) degraded phospholipids into PA and was induced by P starvation [85,86]. SnRK1 (Snf1-related kinase 1) inhibited DGAT1 activities by phosphorylating S/T in the SnRK1 targeting motif [23,52,84], and the activity of *Arabidopsis* SnRK1 was reduced and its catalytic subunit AKIN11 was degraded under P starvation [87]. Therefore, the enhancement of TAG accumulation may be related to the higher activities of DGAT1s regulated by the increase in PA and the decrease in SnRK1 activity under P starvation. In addition to TAG, P starvation can also promote the accumulation of DAG, MGDG, DGDG and SQDG by inducing the expression of *NPC4*, *NPC5*, *PLDZ2*, *PAH1*, *PAH2*, *MGD2*, *MGD3*, *DGD1*, *DGD2*, *SQD1* and *SQD2* in *Arabidopsis* seedlings [81,82]. This result suggested that the increase in total fatty acids is a consequence of the accumulation of TAG, DAG, MGDG, DGDG and SQDG in *B. napus* seedlings under P starvation, which may be caused by gene expression changes similar to those in *Arabidopsis.*

Many studies have shown that most microalgae can accumulate high levels of oil under N starvation [75,88,89,90,91,92,93,94]. For plants, N and C are closely coordinated to affect chloroplast lipid metabolism and TAG content [36,95,96,97]. This study found that low N (5 mM) promoted the accumulation of total fatty acids in roots at 1 h to 6 h and 24 h and in stems at 24 h and 48 h, but inhibited the accumulation of total fatty acids in stems at 6 h and in leaves at 48 h (Figure 8B). In previous reports, low N (0.1 mM and 0.65 mM) promoted TAG accumulation, especially 0.1 mM N with 50 mM sucrose, by inducing the expression of *AtDGAT1*, *AtDGAT2*, *AtPDAT1* and *AtOLEOSIN1* in *Arabidopsis* seedlings [36,96]. In this study, low N (5 mM) did not induce the expression of four *BnaDGAT1s*, *BnaA.DGAT2.a*, *BnaC.DGAT2.a* and two *BnaDGAT3s* in roots, stems and leaves; *BnaA.DGAT2.b* in leaves and stems and *BnaC.DGAT2.b* in stems but promoted the expression of *BnaA.DGAT2.b* in roots and *BnaC.DGAT2.b* in leaves and roots (Figure 7). In previous studies, *Arabidopsis* seedlings were cultured under low N (0.1 mM) using Murashige and Skoog (MS) solid medium, while *B. napus* seedlings were cultured under low N (5 mM) in this study using Hoagland solution without carbon sources. In normal 1/2 Hoagland solution, the nitrogen content was 7.5 mM. Therefore, it may not be sufficient to induce the expression of *BnaDGATs* in *B. napus* under 5 mM N (50 times that under 0.1 mM).

In this study, the accumulation of total fatty acids in roots and leaves at 3 h and 6 h and in stems at 1 h to 6 h was reduced but was promoted in stems at 12 h and 24 h using 15% PEG for stress treatment (Figure 8C). Previous studies showed that drought stress reduced phospholipids (PC, PE, PG), glycolipids (MGDG and DGDG) and total fatty acids in *B. napus* and increased neutral lipids (mainly TAG) [98,99,100,101], which is consistent with maize [102], soybean [103] and cotton [104]. Therefore, the changes in total fatty acids in the roots, stems and leaves of *B. napus* seedlings under drought stress were the combined result of a decrease in phospholipids and an increase in neutral lipids. In this study, drought stress for 24 h increased the expression levels of four *BnaDGAT1s* in leaves, *BnaA.DGAT2.a* and *BnaC.DGAT2.a* in stems and leaves; *BnaA.DGAT2.b* and *BnaC.DGAT2.b* in roots; *BnaA.DGAT3* in stems and roots and *BnaC.DGAT3* in roots, but decreased the expression levels of *BnaA.DGAT2.b* in leaves and stems and *BnaC.DGAT3* in leaves (Figure 7). A previous study showed that the expression of *AtDGAT1* was promoted by ABI4 and ABI5 under drought stress [38]. Therefore, it is necessary to further test whether ABI4 and ABI5 participate in the regulation of *BnaDGAT* expression in *B. napus* under drought stress. Furthermore, drought-inducible elements (MBS) were found in the potential promoters of *BnaA.DGAT2.a*, *BnaA.DGAT2.b*, *BnaC.DGAT2.b*, *BnaA.DGAT3* and *BnaC.DGAT3* (Figure 9 and Appendix A). In *Arabidopsis*, MBS is generally present in the promoter of drought-inducible genes and bound by AtMYB2 in response to drought stress [105,106,107]. The expression of AtMYB2 was induced by drought, salt or ABA treatment in *Arabidopsis* seedlings [105]. Therefore, the expression levels of *BnaA.DGAT2.a*, *BnaA.DGAT2.b*, *BnaC.DGAT2.b*, *BnaA.DGAT3* and *BnaC.DGAT3* may be regulated by BnaMYB2 under drought stress, which needs to be further tested.

For most microalgae, the appropriate salinity (20–40 g/L, i.e., 340–680 mM) can increase the accumulation of lipids, while lipid accumulation will be negatively affected when the salinity is excessive [75]. In *Arabidopsis*, TAG accumulation and the expression of *DGAT1* were promoted by enhancing the expression of *ABI4* and *ABI5* under salt stress (100 mM NaCl) [38]. In this study, it was found that the accumulation of total fatty acids was reduced in roots at 1 h to 24 h and in stems at 1 h to 6 h and 48 h, while it was elevated in leaves at 6 h and 48 h under salt stress (150 mM NaCl; Figure 8D). Moreover, the expression levels of four *BnaDGAT1s* in stems; *BnaA.DGAT2.a* and *BnaC.DGAT2.a* in roots, stems and leaves and *BnaA.DGAT2.b* in roots were increased, but the expression levels of four *BnaDGAT1s* in roots; *BnaA.DGAT2.b* in leaves and stems; *BnaC.DGAT2.b* in stems; *BnaA.DGAT3* in stems and leaves and *BnaC.DGAT3* in roots, stems and leaves at 12–48 h of stress decreased (Figure 7). Therefore, it is necessary to further test and explore the relationship between the accumulation of fatty acids and the expression of *BnaDGATs* in *B. napus* under salt stress.

## 4. Materials and Methods

### 4.1. Identification of DGAT Family Members in B. napus and in Other Plants

To identify candidate DGAT family members in *B. napus*, the CDS and peptide sequences of the three AtDGATs from the *A. thaliana* genome database (http://www.arabidopsis.org/ (accessed on 25 March 2020)) with corresponding Gene IDs (At2G19450, At3G51520 and At1G48300) were retrieved and used as queries to perform BLAST searches in two *B. napus* Genome Databases (BnPIR, http://cbi.hzau.edu.cn/bnapus (accessed on 2 March 2021), and GENOSCOPE, https://www.genoscope.cns.fr/brassicanapus/ (accessed on 25 March 2020)) with the default parameters. The CDSs and genomic DNAs of *BnaDGATs* were cloned from *B. napus* ZS11 using primers (Appendix A), designed according to the nucleotide sequences of *BnaDGATs* retrieved from the two *B. napus* genome databases. The DGATs of *B. oleracea*, *B. rapa*, *B. juncea* and *B. nigra* were downloaded from the *Brassica* Database (BRAD, http://brassicadb.cn/#/ (accessed on 11 October 2021)). *B. carinata* DGATs were derived from the *Brassica* Genomics Database (BGD, http://brassicadb.bio2db.com/ (accessed on 11 October 2021)). DGAT family members in *A. hypogaea*, *G. max*, *R. communis*, *O. sativa*, *Z. mays*, *S. italica*, *M. truncatula*, *B. distachyon* and *J. curcas* were blasted and selected from their corresponding genome databases (Appendix A). Then, the theoretical molecular weight (MW) and isoelectric point (pI) were predicted using the ProtParam tool (https://web.expasy.org/protparam/ (accessed on 22 October 2021)) on the basis of their amino acid sequences. The subcellular location pattern of each BnaDGAT was evaluated via ProtComp v.9.0 in softberry (http://linux1.softberry.com/ (accessed on 22 October 2021)). Multiple sequence alignments of DGATs of *B. napus*, *B. oleracea* and *B. rapa* were performed based on full-length proteins and full-length CDSs by ClustalW, and their identities were evaluated by Sequence Distances in MegAlign software.

### 4.2. Chromosomal Location and Gene Synteny Analysis

The detailed chromosome locations of *DGATs* in *B. napus*, *B. rapa* and *B. oleracea* were acquired from the GFF genome files downloaded from the *B. napus* genomic database (BnPIR, http://cbi.hzau.edu.cn/bnapus (accessed on 11 October 2021)) and *Brassica* Database (BRAD, http://brassicadb.cn/#/ (accessed on 11 October 2021)), respectively, and the predicted locations on the chromosomes were mapped by using TBtools software [62], with red-colored gene names indicated as relative positions. Gene synteny analyses of *DGATs* in *A. thaliana*, *B. oleracea*, *B. rapa*, *B. napus* and *B. nigra* were carried out by using TBtools software [62]. In addition, we calculated the synonymous (Ks), nonsynonymous mutations (Ka) and Ka/Ks ratio at each codon by Tbtools [62]. In addition, the *DGAT* gene pair duplication time was presumed using the formula T = Ks/2R, with R (1.5 × 10^−8^) representing neutral substitution per site per year [108].

### 4.3. Phylogenetic, Gene Structure, Conserved Domain and Motif Analyses

The MUSCLE program in MEGA11 software was used to perform DGAT alignments with amino acid sequences by default parameters, and then the rectangular phylogenetic tree was constructed using the neighbor-joining (NJ) method with 2000 bootstrap replications. The rectangular phylogenetic tree Newick format was saved for constructing the gene structure, conserved domain and motifs. The genomic and coding sequences of DGAT genes were obtained from their corresponding genome databases (Appendix A) and rendered in the Gene Structure Display Server (GSDS2.0; http://gsds.gao-lab.org/ (accessed on 20 October 2021)) [56] to construct their gene structures. Amino acid sequences of DGATs were submitted to the MEME program (Version 5.4.1, https://meme-suite.org/meme/tools/meme (accessed on 21 October 2021)) with the maximum motif search set to 20 and other parameters set to default to identify the conserved protein motifs [109]. A functional search of the conserved domains was performed using the Conserved Domain Database (CDD) with Batch CD-Search in NCBI. The conserved motifs and domains were visualized by using TBtools software [62]. The putative transmembrane domains of DGATs were predicted using TMHMM Server v. 2.0 (https://services.healthtech.dtu.dk/service.php?TMHMM-2.0 (accessed on 10 April 2021)). Multiple sequence alignments of DGATs were performed based on their full-length proteins by ClustalW in MegAlign software and then crested with CLC Sequence Viewer 6.8 software.

### 4.4. Yeast Expression Vector Construction and Transformation

First, the CDSs of ten *BnaDGATs*, one *AtDGAT1* (as a positive control) and one *eGFP* (as a negative control) were cloned into the entry vector pGWC using the corresponding primers (Appendix A) and then recombined into the Gateway vector pYES-DEST52 (Invitrogen). The constructs were introduced into yeast H1246 (*MATα are1-Δ::HIS3, are2-Δ::LEU2, dga1-Δ::KanMX4, lro1-Δ::TRP1 ADE2*) [59] and INVSc1 (*MATα his3Δ1 leu2 trp1-289 ura3-52/MATα his3Δ1 leu2 trp1-289 ura3-52*), according to the *Yeast Protocols Handbook* from Clontech Laboratories Inc. (Mountain View, CA, USA) Transformants were selected on synthetic complete medium lacking uracil (SC-ura).

### 4.5. Nile Red Staining and Microscopy

Lipid droplets in the transformants of yeast H1246 were stained with Nile Red and then visualized on a Leica TCS SP5 (Leica Microsystems, Wetzlar, Germany) laser scanning confocal microscope, as previously described [92].

### 4.6. Analysis of Fatty Acids in Yeast Transformants and B. napus Seedlings by GC–MS

The transformants of yeast H1246 and INVSc1 first cultured with liquid SC-ura medium containing 2% (*w*/*v*) glucose at 30 °C overnight were then diluted to OD_600_ = 0.1 with liquid SC-ura medium containing 2% (*w*/*v*) galactose and 1% (*w*/*v*) raffinose and shaken at 30 °C and 250 rpm for 72 h. The yeast cells were harvested by centrifugation and dried at 55 °C, and then the pellets were ground to a fine powder. For *B. napus* seedlings, samples of roots, stems and leaves were also ground to a fine powder under liquid nitrogen and then freeze-dried. For fatty acid extraction, 50 mg of yeast powder or 15 mg of freeze-dried powder of every *B. napus* sample was incubated in 3 mL of 7.5% (*w*/*v*) KOH in methanol for saponification at 70 °C for 5 h. Then, the pH was adjusted to 2.0 with HCl, and the fatty acid was subjected to methyl-esterification with 2 mL of 14% (*w*/*v*) boron trifluoride in methanol at 70 °C for 2 h. A phase separation was produced by adding 2 mL of 0.9% (*w*/*v*) NaCl and 4 mL of hexane. The upper phase was dried under a nitrogen gas flow and resuspended in acetic ether. Analysis of fatty acid methyl esters (FAMEs) was performed by GC–MS (GC-QQQ, 7890A-7001B, Agilent Technologies, Santa Clara, CA, USA) equipped with a capillary column (HP-FFAP, 30 mm × 0.25 mm ID, 0.25 μm; Agilent Technologies). Hydrogen was used as the carrier gas at a flow rate of 1.0 mL/min. The injection port, transmission line and ion source temperatures were held at 220 °C, 230 °C and 230 °C, respectively. The temperature of the column oven was programmed from 60 to 180 °C at 10 °C/min, then from 180 to 210 °C at 3 °C/min, and finally from 210 to 220 °C at 5 °C/min and held for 15 min. The FA content was quantified using heptadecanoic acid (C17:0, Sigma, St. Louis, MO, USA) as an internal standard added to samples prior to extraction. All experiments were performed in biological triplicates.

### 4.7. B. napus Seedling Treatments and Sampling

The seedlings of *B. napus* Westar were hydroponically cultured in 1/2 Hoagland solution [110] for six weeks with a 16 h day/8 h night cycle at 23 °C and then were used for the stresses of P starvation (KH_2_PO_4_ replaced by an equimolar amount of KCl), low N (KNO_3_ replaced with an equimolar amount of KCl), drought (15% *w*/*v* PEG6000) and salinity (150 mM NaCl). Seedlings cultured in normal 1/2 Hoagland solution were used as the control group (mock). Roots, stems and leaves of seedlings were sampled at seven time points: 0 h, 1 h, 3 h, 6 h, 12 h, 24 h and 48 h. The collected samples were immediately dipped in liquid nitrogen and then stored at −80 °C for the relative expression levels of *BnaDGATs* and fatty acid analysis. All experiments were performed in biological triplicates.

### 4.8. Genomic DNA Extraction, Total RNA Isolation, Primary cDNA Synthesis and qRT–PCR

Genomic DNA of *B. napus* ZS11 was extracted using a method modified from a CTAB-based protocol [111] for cloning genomic DNA of *BnaDGATs*. Total RNA of *B. napus* was extracted using an RNAprep Pure Plant Kit (DP432, TIANGEN BIOTECH Co., Ltd., Beijing, China), and *B. napus* complementary DNA was synthesized using TransScript One-Step gDNA Removal and cDNA Synthesis SuperMix (AT311-03, TransGen Biotech, Beijing, China), according to the manufacturer’s instructions. For qRT–PCR, 2 μg of total RNA was first used for the synthesis of first-strand cDNA using oligo(dT)_18_ as a primer, and then qRT–PCR was performed on a CFX Connect^TM^ Real-Time PCR system (Bio–Rad, Hercules, CA, USA) using EvaGreen 2× qPCR MasterMix (MasterMix-S, Abm, Vancouver, BC, Canada), according to the manufacturer’s instructions. The specificity of primers (Appendix A) for qRT–PCR was confirmed by separating the products on agarose gels and clone sequencing. *BnaACT7* was used as an internal reference gene, and the relative expression levels of each *BnaDGAT* to *BnaACT7* in leaves, stems and roots at 0 h were set to 1. All qRT-PCRs were performed in biological triplicates.

### 4.9. Expression Pattern Analysis Based on RNA-Seq Datasets

The transcript level was calculated based on publicly released data. RNA-Seq datasets of different tissues at diverse stages of development were obtained from BnTIR (http://yanglab.hzau.edu.cn/BnTIR (accessed on 10 November 2021)) [60] and BrassicaEDB (https://brassica.biodb.org/ (accessed on 10 November 2021)) [61]. The extracted data from BnTIR and BrassicaEDB were normalized by log_2_(TPM) and log_2_(FPKM), respectively, and the heatmaps were generated by TBtools [62].

### 4.10. Analyses of Transcription Factors and miRNAs Targeting BnaDGATs and Cis-Acting Elements in BnaDGAT Promoters

Transcription factors regulating *BnaDGATs* were predicted using PlantRegMap [112], with *B. napus* as the target. The full-length cDNA sequences of *BnaDGAT* homologues were submitted to the psRNATarget website (https://www.zhaolab.org/psRNATarget/ (accessed on 25 November 2021)) [113] for a potential miRNA search against the *B. napus* miRNA database. For *cis*-element analysis, the regions 776–1500 bp upstream of the start codon of *BnaDGATs* and *AtDGATs* were subjected to the plantCARE database (http://bioinformatics.psb.ugent.be/webtools/plantcare/html/ (accessed on 20 September 2021)) [114] for *cis*-element searching.

## 5. Conclusions

In summary, ten BnaDGATs were identified and grouped into three different DGAT subfamilies. These BnaDGATs are derived from three different ancestors and evolved separately during plant evolution, as proposed by analyzing physicochemical properties, chromosome location, gene synteny, phylogenetic tree, exon/intron gene structure, conserved domain and motif compositions and TMDs. BnaDGAT1s possess the ability to introduce the fatty acids C10, C12, C14, C16 and C18 into TAG in *S. cerevisiae* and a higher preference for the fatty acids C16:0, C16:1n7, C18:0 and C18:1n9 than C10:0, C12:0, C14:0, C14:1 and C18:1n7. BnaDGAT1s are the main diacylglycerol acyltransferases that synthesize TAGs in *B. napus*. The role of BnaDGAT2s and BnaDGAT3s in TAG synthesis in *B. napus* needs to be further clarified. Some B3, bZIP, MYB-like transcription factors and other transcription factors involved in the response to light signals may regulate the expression of *BnaDGATs*. P starvation increased fatty acid accumulation in *B. napus* seedlings. The relationships between the expression of *BnaDGATs* and the accumulation of lipids in *B. napus* under low N, drought and salt conditions remain to be further confirmed. Overall, the findings of this study contribute to the understanding of BnaDGAT genes in fatty acid biosynthesis and abiotic stress responses in oilseed rape.

## Figures and Tables

**Figure 1 plants-11-01156-f001:**
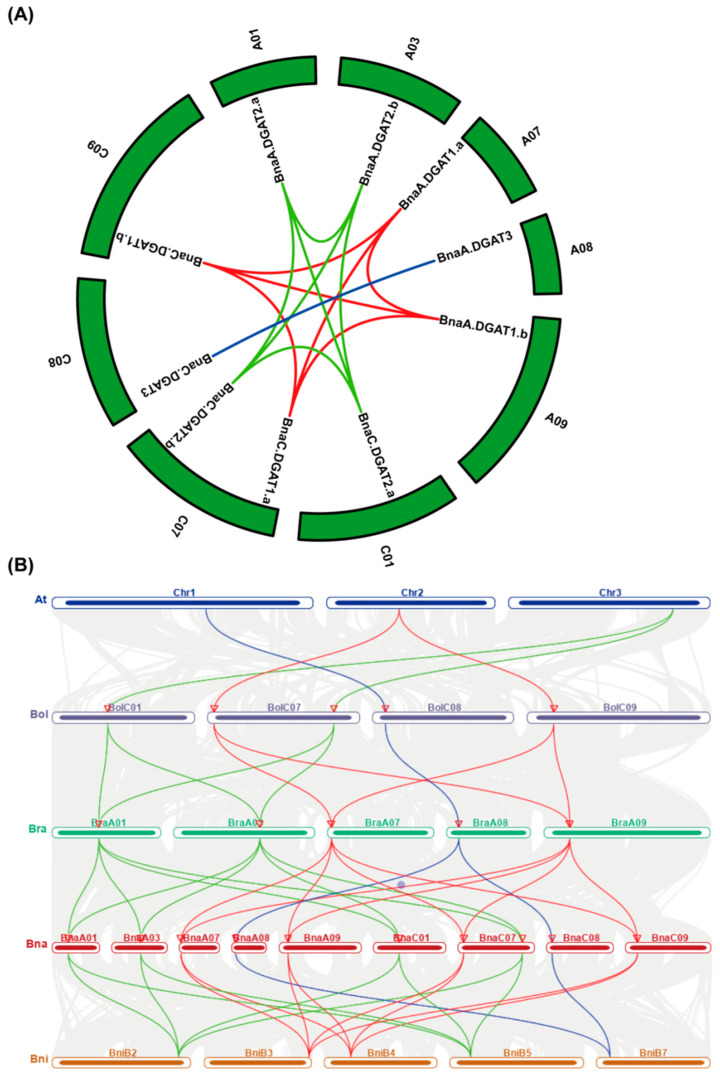
Synteny analysis of *DGATs* in *Arabidopsis thaliana*, *Brassica oleracea*, *Brassica rapa*, *Brassica napus* and *Brassica nigra*. (**A**) Synteny analysis of the *BnaDGAT* family in *B. napus*. Red-, green- and blue-colored lines indicate the *BnaDGAT1* subfamily, *BnaDGAT2* subfamily and *BnaDGAT3* subfamily genes, respectively; (**B**) Synteny analysis of *DGATs* between *A. thaliana*, *B. oleracea*, *B. rapa*, *B. napus* and *B. nigra*. Red, green and blue lines indicate the syntenic *DGAT1*, *DGAT2* and *DGAT3* gene pairs between the denoted species, respectively. In the background, the grey line represents collinear blocks.

**Figure 2 plants-11-01156-f002:**
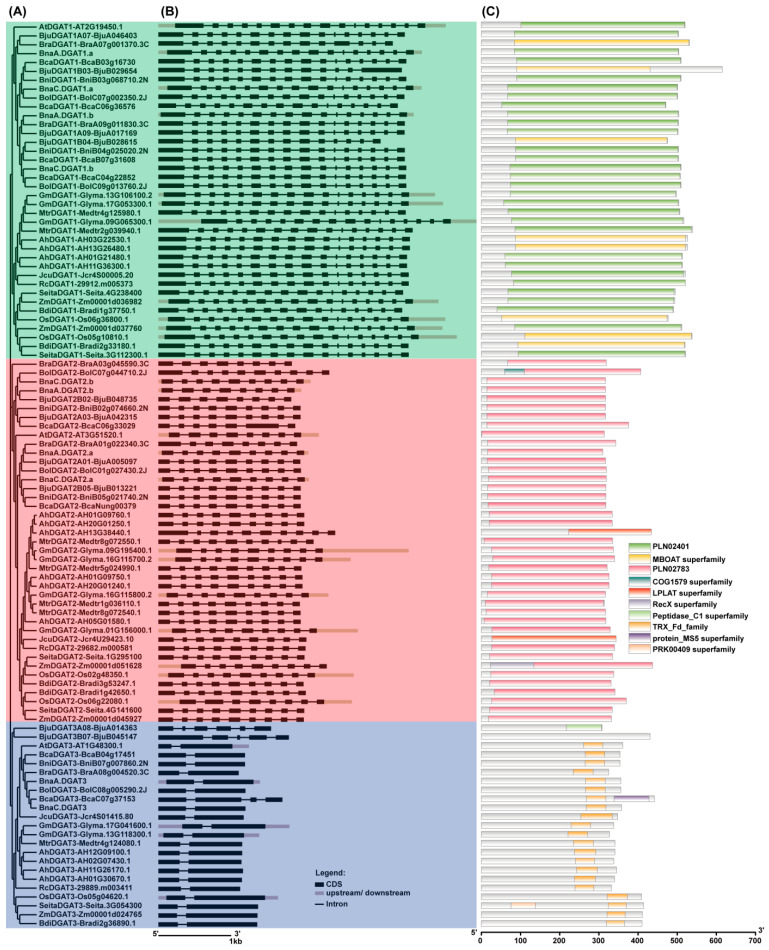
Analyses of phylogenetic tree, gene structures and conserved domains of DGAT family members in *Brassica napus* and other plant species. (**A**) Phylogenetic tree. DGAT protein sequences were used to construct the phylogenetic tree by MEGA11 software with the neighbor-joining method and 2000 bootstrap replications; (**B**) Gene structures. The genomic DNA and CDSs of DGATs were used to construct the gene structures by the Gene Structure Display Serve (GSDS2.0; http://gsds.gao-lab.org/ (accessed on 20 October 2021)) [56]. Black boxes denote exons within coding regions, and the lines connecting exons represent introns. Grey boxes indicate the upstream or downstream regions of the CDS of DGAT genes. The length of the boxes represents the size of the corresponding exons; (**C**) Conserved domains. A functional search of the conserved domains was performed using the Conserved Domain Database (CDD) with Batch CD-Search in NCBI. *BnaDGAT1s*, *BnaDGAT2s* and *BnaDGAT3s* were cloned from *B. napus* ZS11. The protein, genomic DNA and CDSs of DGAT family members in *B. oleracea*, *B. rapa*, *B. nigra*, *B. juncea*, *B. carinata, A. thaliana*, *A. hypogaea*, *G. max*, *R. communis*, *O. sativa*, *Z. mays*, *S. italica*, *M. truncatula*, *B. distachyon* and *J. curcas* were blasted and selected from their corresponding genome databases (Appendix A). All DGAT1s contained the PLN02401 domain (diacylglycerol o-acyltransferase) or the MBOAT domain (membrane-bound O-acyltransferase family). All DGAT2 homologues shared the PLN02783 domain (diacylglycerol o-acyltransferase), except AH13G38440.1 and Jcr4U29423.10, which had an LPLAT superfamily domain. All DGAT3 homologues shared the TRX_Fd_family domain, except BjuB045147 and BjuA014363.

**Figure 3 plants-11-01156-f003:**
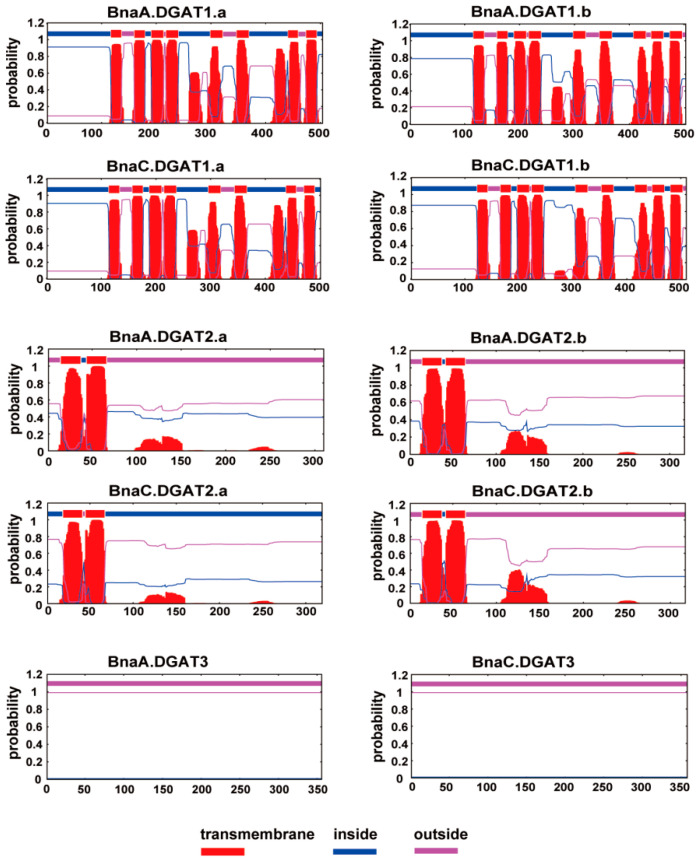
Predicted transmembrane domains (TMDs) of BnaDGATs. Each BnaDGAT1s had eight or nine putative TMDs, and each BnaDGAT2s showed two putative TMDs. However, neither BnaDGAT3 had putative TMDs. The putative TMDs of BnaDGATs were predicted using TMHMM Server v. 2.0 (https://services.healthtech.dtu.dk/service.php?TMHMM-2.0 (accessed on 10 April 2021)). Regions of BnaDGATs predicted to be located inside or outside the membrane are shown in bold blue and pink lines, respectively.

**Figure 4 plants-11-01156-f004:**
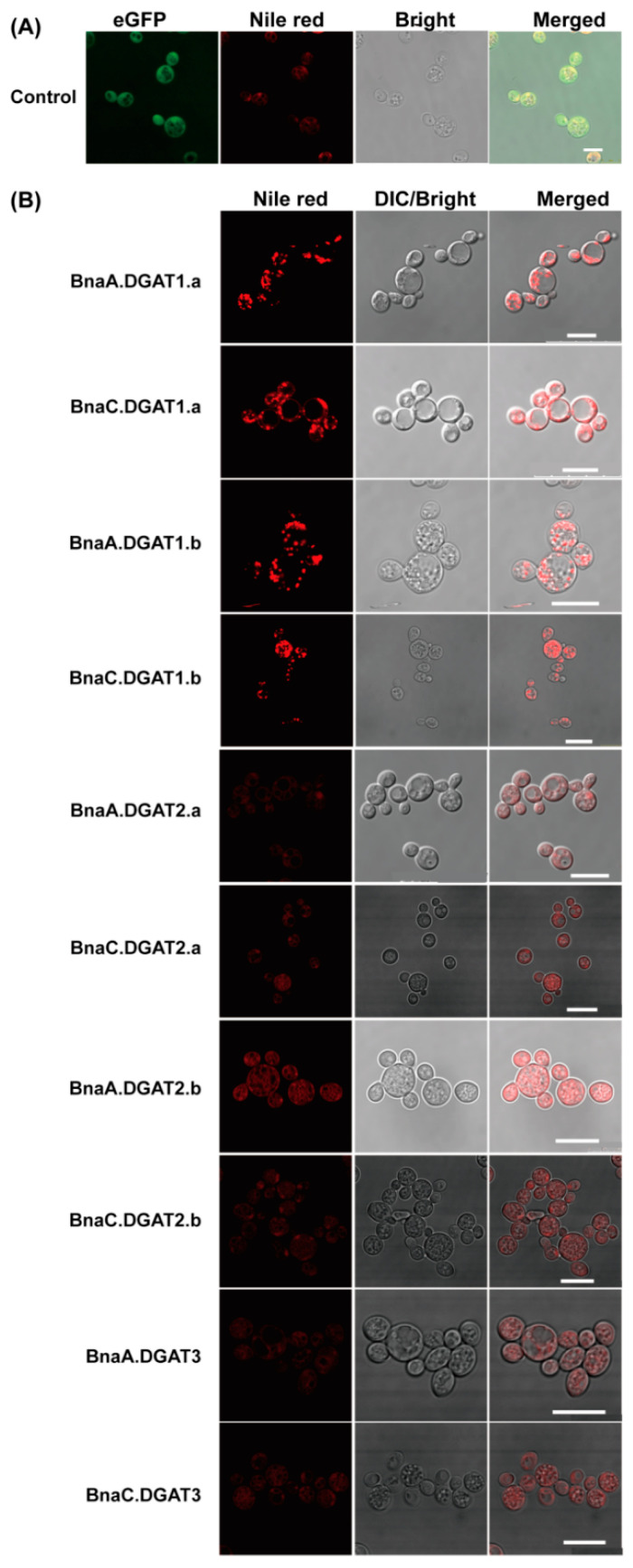
Oil bodies in *S. cerevisiae* H1246 expressing *BnaDGATs* by Nile red staining. (**A**) Negative control. Yeast H1246 expressing *eGFP*; (**B**) Yeast H1246 expressing different *BnaDGATs*. Oil droplets were observed in yeast H1246 expressing *BnaDGAT1s* but were not detectable and/or were very weak in yeast H1246 expressing *BnaDGAT2s* and *BnaDGAT3s*. Bar = 10 μm.

**Figure 5 plants-11-01156-f005:**
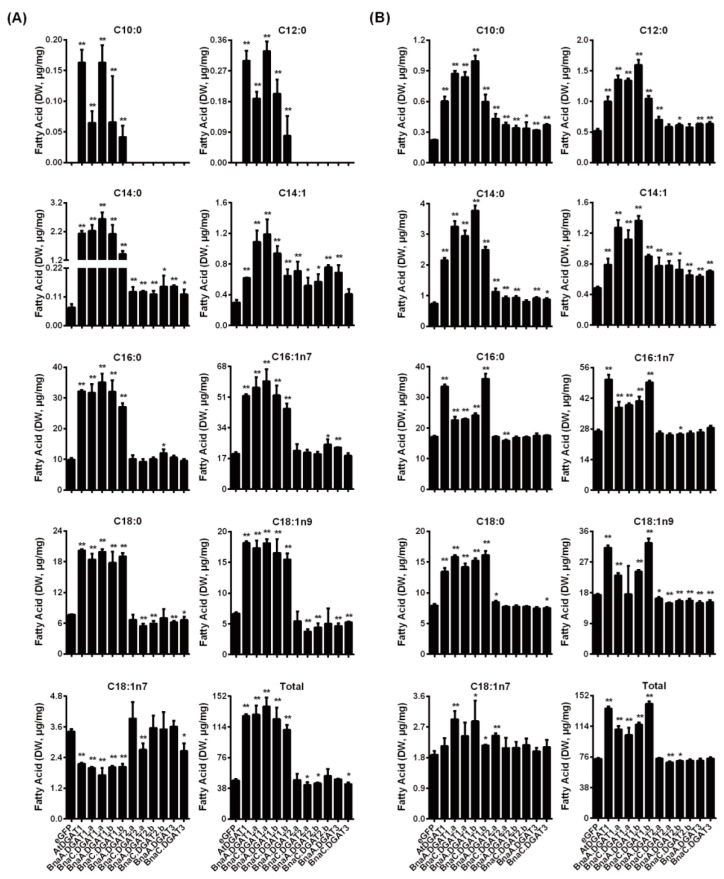
Analysis of the fatty acid compositions and content in yeast H1246 and INVSc1 expressing different *BnaDGATs*. (**A**) Yeast H1246; (**B**) Yeast INVSc1. Yeast expressing *eGFP* was the negative control, and that expressing *AtDGAT1* was the positive control. Data represent averages of three replicates ± SD. Statistical analysis was carried out by Student’s *t* test. Asterisks indicate the fatty acids with a significant difference from those in the negative control (* *p* < 0.05; ** *p* < 0.01).

**Figure 6 plants-11-01156-f006:**
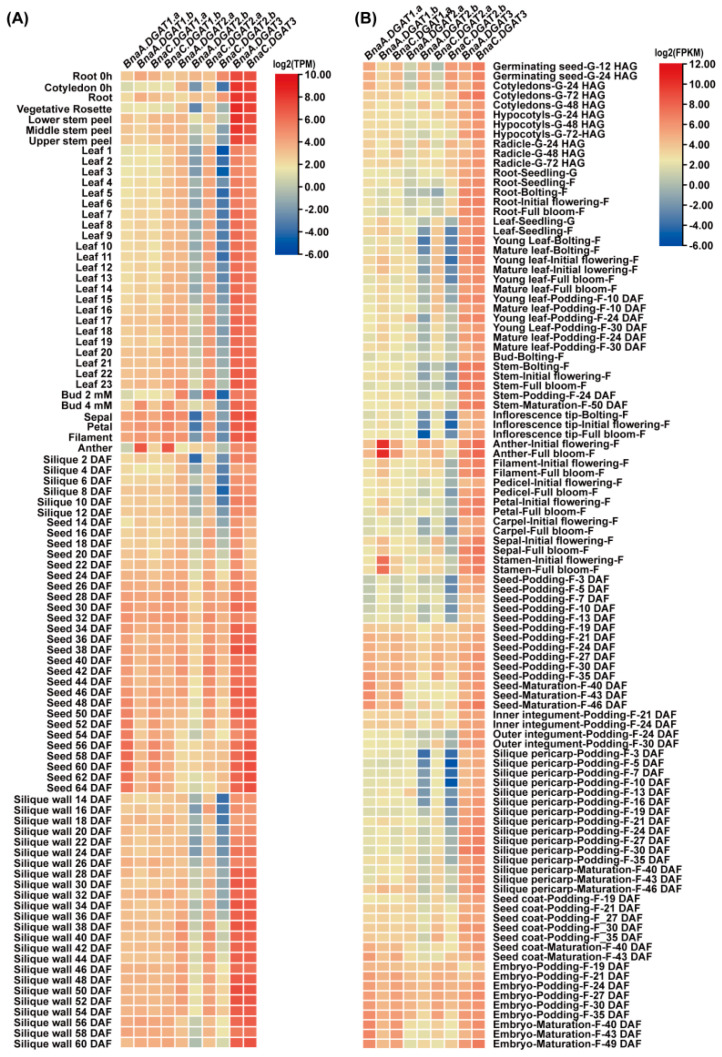
Transcriptomic analysis of *BnaDGATs* in different tissues. (**A**) The analysis based on BnTIR; (**B**) The analysis based on BrassicaEDB. RNA-Seq datasets of different tissues at diverse stages of development were obtained from BnTIR (http://yanglab.hzau.edu.cn/BnTIR (accessed on 10 November 2021)) [60] and BrassicaEDB (https://brassica.biodb.org/ (accessed on 10 November 2021)) [61]. The extracted data from BnTIR and BrassicaEDB were normalized by log_2_(TPM) and log_2_(FPKM), respectively, and the heatmaps were generated by TBtools [62].

**Figure 7 plants-11-01156-f007:**
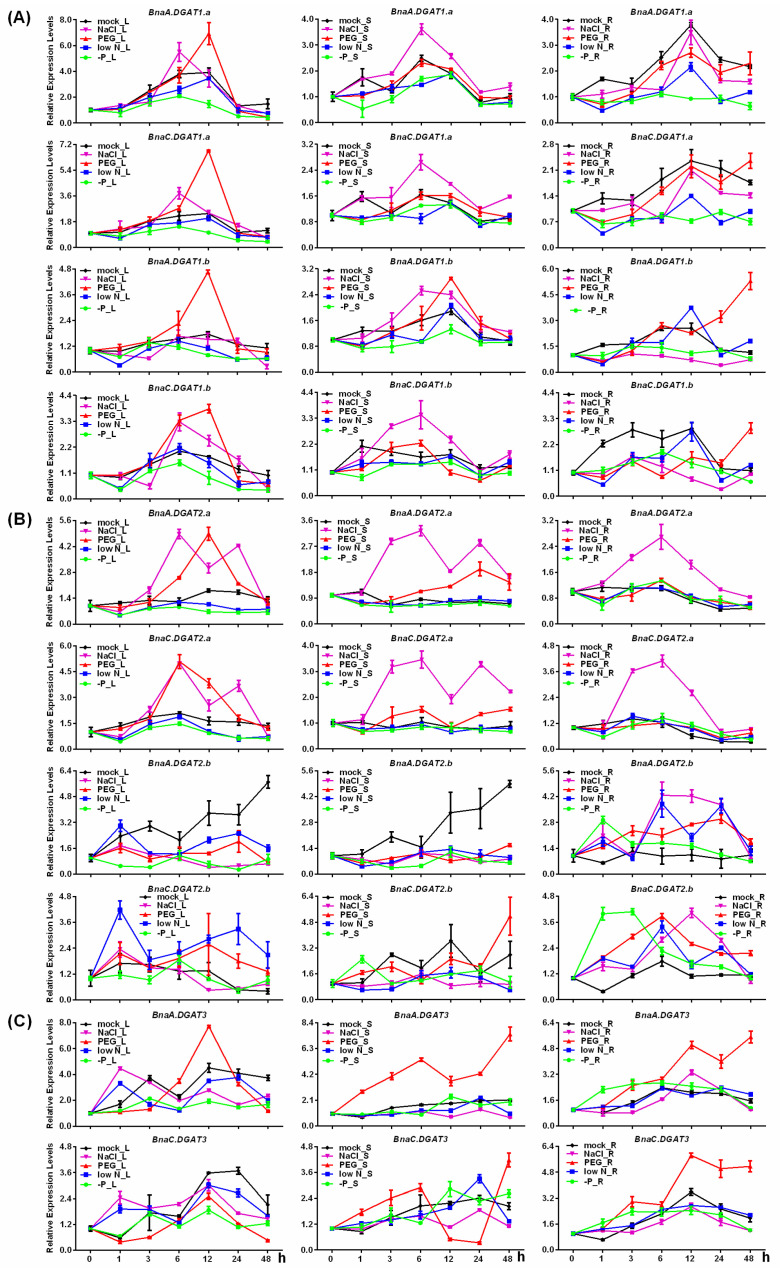
qRT–PCR analyses of different *BnaDGATs* under different stresses. (**A**) qRT–PCR analyses of *BnaDGAT1s* under different stresses; (**B**) qRT–PCR analyses of *BnaDGAT2s* under different stresses; (**C**) qRT–PCR analyses of *BnaDGAT3s* under different stresses. For dehydration and salt stress treatments, six-week-old seedlings of *B. napus* were grown hydroponically in solution containing 15% (*w*/*v*) PEG6000 and 150 mM NaCl plus 1/2 Hoagland solution. In the Pi-deficient treatment (-P), KH_2_PO4 in 1/2 Hoagland solution was replaced by equimolar amounts of KCl. In the low nitrogen treatment (low N), KNO_3_ in 1/2 Hoagland solution was replaced by equimolar amounts of KCl. Seedlings growing in 1/2 Hoagland solution for various periods of time were used as the mock. Tissues at 0 h, 1 h, 3 h, 6 h, 12 h, 24 h and 48 h were sampled. *BnaACT7* was used as an internal control. The expression level of each gene at 0 h in leaves (L), stems (S) and roots (R) was arbitrarily set as 1. The data represent averages of three replicates ± SD.

**Figure 8 plants-11-01156-f008:**
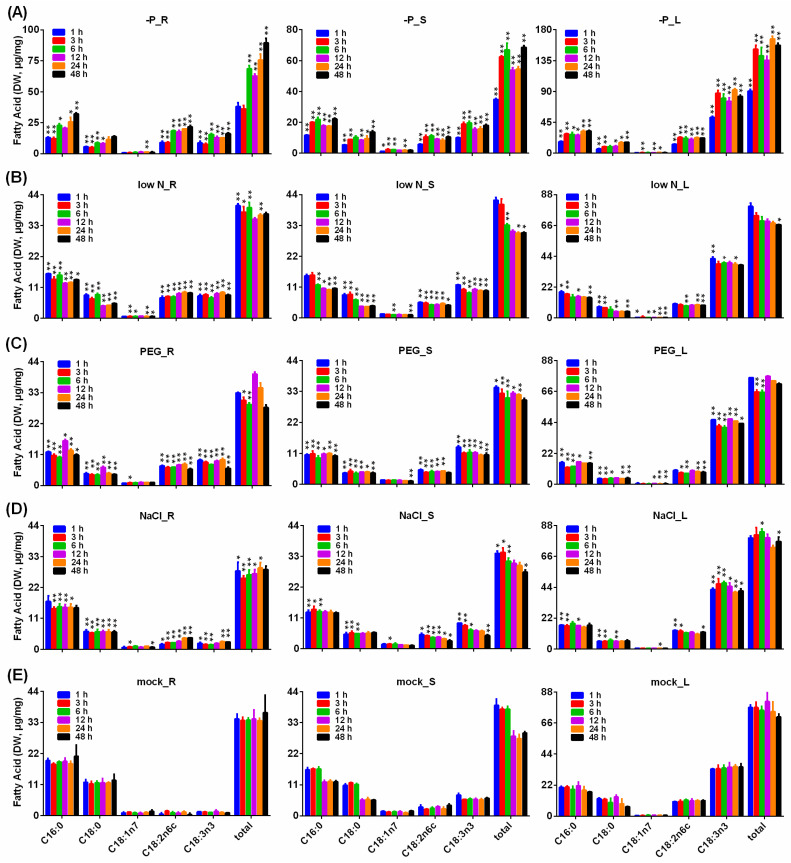
Main fatty acid content in roots, stems and leaves under abiotic stresses. (**A**) Pi-deficient treatment (-P); (**B**) Low nitrogen treatment (low N). (**C**) Dehydration treatment (PEG); (**D**) Salt treatment (NaCl); (**E**) Mock. For dehydration and salt stress treatments, six-week-old seedlings of *B. napus* were grown hydroponically in solution containing 15% (*w*/*v*) PEG6000 and 150 mM NaCl plus 1/2 Hoagland solution. In the Pi-deficient treatment, KH_2_PO_4_ in 1/2 Hoagland solution was replaced by equimolar amounts of KCl. In the low nitrogen treatment, KNO_3_ in 1/2 Hoagland solution was replaced by equimolar amounts of KCl. Seedlings grown in 1/2 Hoagland solution for various periods of time were used as the mock. The FA content was quantified by GC–MS with heptadecanoic acid (C17:0) as an internal standard. The data represent averages of three replicates ± SD. Asterisks indicate the fatty acid content with a significant difference from those in the mock (* *p* < 0.05; ** *p* < 0.01).

**Figure 9 plants-11-01156-f009:**
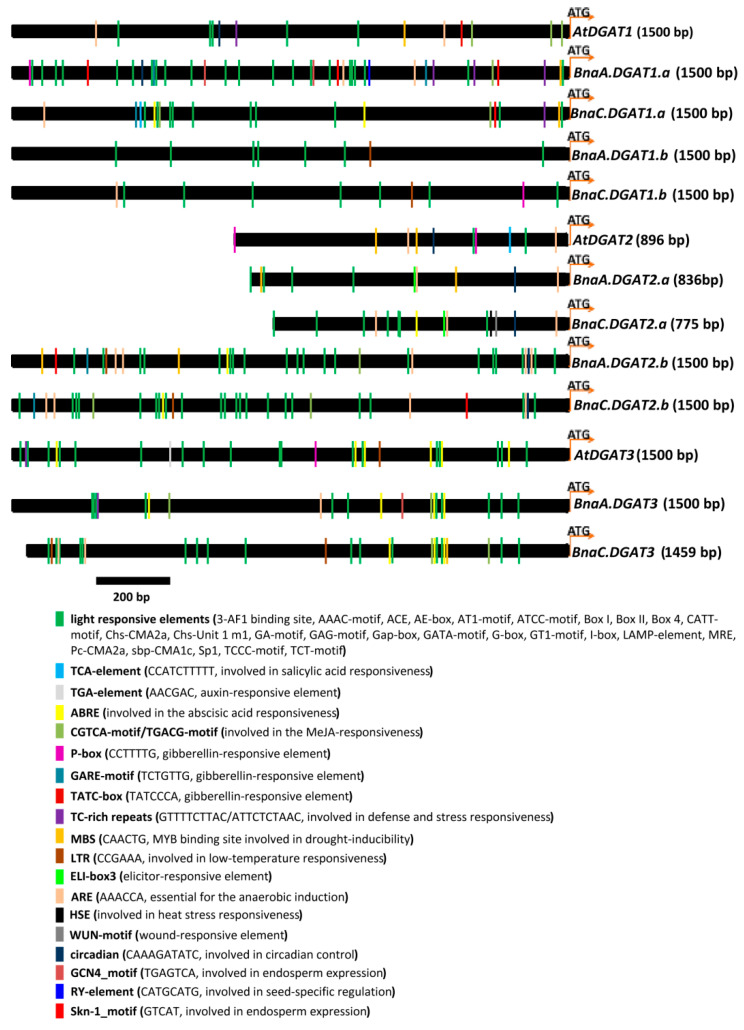
Potential *cis*-acting regulatory elements upstream of the start codons of *BnaDGATs* and *AtDGATs* deduced by PlantCARE. *P_BnaA.DGAT1.a_*, *P_BnaC.DGAT1.a_*, *P_BnaA.DGAT1.b_*, *P_BnaA.DGAT2.a_*, *P_BnaC.DGAT2.a_*, *P_BnaA.DGAT2.b_*, *P_BnaC.DGAT2.b_*, *P_BnaA.DGAT3_* and *P_BnaC.DGAT3_* were cloned from *B. napus* ZS11. *P_AtDGAT1_*, *P_AtDGAT2_* and *P_AtDGAT3_* were derived from the Arabidopsis database (https://www.arabidopsis.org (accessed on 25 March 2020)). *P_BnaC.DGAT1.b_* was extracted from the ZS11 genome database (https://www.ncbi.nlm.nih.gov/assembly/GCF_000686985.2/ (accessed on 20 September 2021)). Black bar = 200 bp.

**Table 1 plants-11-01156-t001:** Members of the DGAT family in *A. thaliana* and six *Brassica* species forming U’s triangle according to their genome databases.

*Arabidopsis thaliana*	*Brassica napus*		*Brassica rapa*	*Brassica oleracea*	*Brassica nigra*	*Brassica juncea*	*Brassica carinata*
	Brassica_napus_v4.1	ZS11	Brara_Chiifu_V3.0	Braol_JZS_V2.0	Brani_Ni100_V2	Braju_tum_V1.5	
AtDGAT1 At2G19450	BnaA07g36000D	BnaA07G0011800ZS	BraA07g001370.3C	—	—	BjuA046403-A07	—
	BnaAnng30990D	BnaA09G0121200ZS	BraA09g011830.3C	—	—	BjuA017169-A09	—
	—	—	—	—	BniB03g068710.2N	BjuB029654-B03	BcaB03g16730
	—	—	—	—	BniB04g025020.2N	BjuB028615-B04	BcaB07g31608
	BnaCnng52810D	BnaC07G0026200ZS	—	BolC07g002350.2J	—	—	BcaC04g22852
	—	BnaC09G0126800ZS	—	BolC09g013760.2J	—	—	BcaC06g36576
AtDGAT2 At3G51520	BnaA01g19390D	BnaA01G0206700ZS	BraA01g022340.3C	—	—	BjuA005097-A01	—
	BnaA03g41350D	BnaA03G0420700ZS	BraA03g045590.3C	—	—	BjuA042315-A03	—
	—	—	—	—	BniB02g074660.2N	BjuB048735-B02	BcaC06g33029
	—	—	—	—	BniB05g021740.2N	BjuB013221-B05	BcaNung00379
	BnaC01g23350D	BnaC01G0259700ZS	—	BolC01g027430.2J	—	—	—
	BnaC07g32270D	BnaC07G0393500ZS	—	BolC07g044710.2J	—	—	—
AtDGAT3 At1G48300	BnaA08g03400D	BnaA08G0039500ZS	BraA08g004520.3C	—	—	BjuA014363-A08	—
	—	—	—	—	BniB07g007860.2N	BjuB045147-B07	BcaB04g17451
	BnaC08g46660D	BnaC08G0049200ZS	—	BolC08g005290.2J	—	—	BcaC07g37153

AtDGATs were extracted from the *Arabidopsis* database (TAIR, http://www.arabidopsis.org/ (accessed on 25 March 2020)); BnaDGATs in Brassica_napus_v4.1 were selected from *Brassica napus* Genome Browser (http://www.genoscope.cns.fr/brassicanapus/ (accessed on 25 March 2020)); ZS11 indicates the gene locus IDs of BnaDGATs in *B. napus* ZS11 genome database (BnPIR; http://cbi.hzau.edu.cn/bnapus/index.php (accessed on 2 March 2021)); *B. rapa*, *B. oleracea*, *B. nigra* and *B. juncea* DGATs were blasted in the *B**rassica* database (BRAD, http://brassicadb.cn/#/ (accessed on 11 October 2021)); *B. carinata* DGATs were derived from the Brassica Genomics Database (BGD, http://brassicadb.bio2db.com/ (accessed on 11 October 2021)). The concept of U’s triangle described natural allopolyploidization events in *Brassica* using three diploids (*B. rapa* (A genome), *B. nigra* (B), and *B. oleracea* (C) and three derived allotetraploids (*B. juncea* (AB genome), *B. napus* (AC), and *B. carinata* (BC)) [51].

**Table 2 plants-11-01156-t002:** Cloning and characterization of the members of *BnaDGAT* family.

Scheme	Gene Name	Gene Locus ID	Chromosome Location	Protein				Putative Promoter
				AA	pI	Mw (Da)	Subcellular Location	
*DGAT1*	*BnaA.DGAT1.a*	BnaA07G0011800ZS	A07:935,937-939,473	504	8.75	57,743.80	E.R.	1759 bp
	*BnaA.DGAT1.b*	BnaA09G0121200ZS	A09:7,271,130-7,274,477	503	8.09	57,241.36	E.R.	1488 bp
	*BnaC.DGAT1.a*	BnaC07G0026200ZS	C07:4,705,347-4,708,653	501	8.43	57,538.87	E.R.	1657 bp
	*BnaC.DGAT1.b*	BnaC09G0126800ZS	C09:9,270,137-9,273,644	510	8.20	57,958.08	E.R.	1500 bp
*DGAT2*	*BnaA.DGAT2.a*	BnaA01G0206700ZS	A01:12,907,146-12,908,853	368	8.77	41,813.30	E.R.	836 bp
	*BnaA.DGAT2.b*	BnaA03G0420700ZS	A03:22,826,130-22,827,759	367	8.32	41,280.40	E.R.	775 bp
	*BnaC.DGAT2.a*	BnaC01G0259700ZS	C01:20,695,151-20,696,870	319	8.88	36,103.24	E.R.	2151 bp
	*BnaC.DGAT2.b*	BnaC07G0393500ZS	C07:51,188,778-51,190,546	317	7.75	35,636.69	E.R.	1973 bp
*DGAT3*	*BnaA.DGAT3*	BnaA08G0039500ZS	A08:3,290,776-3,291,925	356	8.90	38,476.61	None	1660 bp
	*BnaA.DGAT3*	BnaC08G0049200ZS	C08:4,709,675-4,710,823	356	8.84	38,184.10	None	1459 bp

Gene Name shows the names of BnaDGATs annotated according to the nomenclature principles of *Brassica* genes (genus-species-genome-gene name-locus-allele) [54]. The gene locus IDs, chromosome locations and orientation of *BnaDGATs* were derived from the *B. napus* ZS11 genome database (BnPIR; http://cbi.hzau.edu.cn/bnapus/index.php (accessed on 2 March 2021)). The isoelectric point (pI) and molecular weight (Mw) of BnaDGAT proteins were predicted using the ProtParam tool (https://web.expasy.org/protparam/ (accessed on 22 October 2021)). Subcellular localization patterns of BnaDGAT1s and BnaDGAT2s were predicted, and the results showed that they were located in the endoplasmic reticulum (E.R.), whereas no BnaDGAT3s were evaluated using ProtComp v.9.0 in softberry (http://linux1.softberry.com/ (accessed on 22 October 2021)).

**Table 3 plants-11-01156-t003:** Ks/Ka values between *BnaDGAT* and their paralogous gene pairs in *Brassica napus* and their orthologous gene pairs in *Arabidopsis*.

	Gene_1	Gene_2	Ka	Ks	Ka/Ks	Duplication Time (MYA)	Average (MYA)
Orthologous gene pairs	*AtDGAT1*	*BnaA.DGAT1.a*	0.062	0.320	0.194	10.67	15.06
	*AtDGAT1*	*BnaA.DGAT1.b*	0.064	0.400	0.161	13.33	
	*AtDGAT1*	*BnaC.DGAT1.a*	0.064	0.347	0.184	11.56	
	*AtDGAT1*	*BnaC.DGAT1.b*	0.060	0.415	0.145	13.82	
	*AtDGAT2*	*BnaA.DGAT2.a*	0.098	0.486	0.201	16.21	
	*AtDGAT2*	*BnaA.DGAT2.b*	0.123	0.448	0.275	14.92	
	*AtDGAT2*	*BnaC.DGAT2.a*	0.099	0.495	0.200	16.51	
	*AtDGAT2*	*BnaC.DGAT2.b*	0.124	0.436	0.284	14.55	
	*AtDGAT3*	*BnaA.DGAT3*	0.081	0.568	0.143	18.92	
	*AtDGAT3*	*BnaC.DGAT3*	0.086	0.603	0.142	20.11	
Paralogous gene pairs	*BnaA.DGAT1.a*	*BnaA.DGAT1.b*	0.056	0.346	0.161	11.52	8.36
	*BnaA.DGAT1.a*	*BnaC.DGAT1.a*	0.003	0.041	0.064	1.37	
	*BnaA.DGAT1.a*	*BnaC.DGAT1.b*	0.054	0.329	0.163	10.97	
	*BnaA.DGAT1.b*	*BnaC.DGAT1.a*	0.058	0.358	0.162	11.92	
	*BnaA.DGAT1.b*	*BnaC.DGAT1.b*	0.007	0.097	0.073	3.22	
	*BnaC.DGAT1.a*	*BnaC.DGAT1.b*	0.056	0.328	0.170	10.95	
	*BnaA.DGAT2.a*	*BnaA.DGAT2.b*	0.111	0.347	0.321	11.58	
	*BnaA.DGAT2.a*	*BnaC.DGAT2.a*	0.009	0.085	0.107	2.83	
	*BnaA.DGAT2.a*	*BnaC.DGAT2.b*	0.112	0.372	0.301	12.41	
	*BnaA.DGAT2.b*	*BnaC.DGAT2.a*	0.111	0.376	0.296	12.52	
	*BnaA.DGAT2.b*	*BnaC.DGAT2.b*	0.010	0.092	0.107	3.07	
	*BnaC.DGAT2.a*	*BnaC.DGAT2.b*	0.111	0.383	0.290	12.78	
	*BnaA.DGAT3*	*BnaC.DGAT3*	0.019	0.107	0.179	3.57	

## Data Availability

Not applicable.

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
