# Peer review of "Genome-Wide Characterization of DGATs and Their Expression Diversity Analysis in Response to Abiotic Stresses in Brassica napus"

_plants, 2022, doi:10.3390/plants11091156_

Round 1
Reviewer 1 Report
The article entitled "Genome-wide Characterization of Tetraploid Brassica napus DGATs and Their Expression Diversity Analysis in Response to Abiotic Stresses" by Yin et al is very well drafted.
DGAT is one of the highly studied gene family in Brassica including napus. Therefore the section about the phylogenetic distribution, intron-exon structure and transcriptome profiling do not have any novelty. Similarly, the prediction of transmembrane domain and cis-elements do not have relevance.
The strength (somewhat) is the functional characterization of DEGAT homologs.
To strengthen the understanding of DGAT, authors need to provide more data related to genetic variation and the amino acids having a significant role in the functionality. Only a few DGAT showed the desired function in the yeast system but the question remains on the inability of other DGATs.
Authors need to remove the enlisting all the gene names to avoid redundancy with the information provided in the tables and text.
Reviewer 2 Report
Present title need improvments -"Genome-wide Characterization of Tetraploid Brassica napus DGATs and Their Expression Diversity Analysis in Response to Abiotic Stresses" - here "Tetraploid" is not required, similarly better to move the species name at the end.
Abstract
"The analysis of the putative promoters of BnaDGATs showed that there are many potential cis-elements involved in response to light, methyl jasmonate, gibberellin, drought, salicylic acid, defence and stresses." - Authors can remove this from the abstract since predicted cis-elements have very less reliability due to small motif size which has a very high chance of random occurrence in the DNA. The authors can find the exactly similar frequency of these motifs in introns or intergenic regions.
"that P starvation can promote the accumulation of fatty acids" - Authors can better connect this with the rest of the abstract. At present, it looks standalone.
Introduction
DGAT homologs have been extensively studied in Brassica napus. The authors can find several papers describing expression dynamics and functional characterization of DGAT homologs in Brassica napus. Authors seem to ignore such studies.
Results
Table 2 - here the orientation of genes like forward or Reverse does not make any sense. I suggest removing this column. Similarly, keep either CDS or AA.
2.2 Chromosomal Location and Collinearity Analysis - 1st paragraph in this section is unnecessary descriptive. can be condensed down to 2-3 sentences. Similarly, better to move Figure 1 to supplementary. Most of the information is already provided in the tables and other figures.
2.3. Evolutionary Relationship and Exon/Intron Gene Structure Analysis of BnaDGATs
this section is unnecessary descriptive. can be condensed down to 2-3 sentences.
Round 2
Reviewer 1 Report
The authors have addressed most of my concerns. There are still several typos and grammatical issues. for instance "Thra total of ough GBS analysis, 5.662 Gb of clean data was generated for 21 samples" at line 194.
Author Response
We reexamined carefully our manuscript and could not find errors. The sentence "Thra total of ough GBS analysis, 5.662 Gb of clean data was generated for 21 samples" was not find at line 194 and anywhere in our manuscript. This comment may have been misplaced for revisions to other people's articles here.
Reviewer 2 Report
The authors have addressed all of my concerns. The revised version of the MS looks improved as expected.
Author Response
Thank you very much.